



**Sigmoidal Water Retention Function with Improved Behavior in Dry**
**and Wet Soils**
Gerrit H. de Rooij, Juliane Mai, and Raneem Madi
G.H. de Rooij and R. Madi, Helmholtz Centre for Environmental Research – UFZ GmbH,  Soil
System Science Dept., Theodor–Lieser–Strasse 4, 06120 Halle (Saale), Germany; R. Madi,
current address: GFI Grundwasser–Consulting–Institut GmbH, Meraner Strasse 10, 01217
Dresden, Germany.
J. Mai, University of Waterloo, Dept. Civil and Environmental Engineering, 200 University
Ave West, Waterloo, ON N2L 3G1, Canada.
Corresponding author: G.H. de Rooij (gerrit.derooij@ufz.de)





## Abstract

A popular parameterized soil water retention curve (SWRC) has a hydraulic conductivity
curve associated with it that can have an infinite slope at saturation. The problem was
eliminated before by giving the SWRC a non–zero air–entry value. This improved version
still has an asymptote at the dry end, which limits its usefulness for dry conditions and
causes its integral to diverge for commonly occurring parameter values. We therefore
joined the parameterizations' sigmoid mid–section to a logarithmic dry section ending at
zero water content for a finite matric potential, as was done previously for a power–law
type SWRC. We selected five SWRC parameterizations that had been proven to produce
unproblematic near–saturation conductivities and fitted these and our new curve to data
from 21 soils. The logarithmic dry branch gave more realistic extrapolations into the dry
end of both the retention and the conductivity curves than an asymptotic dry branch. We
tested the original curve, its first improvement, and our second improvement by feeding
them into a numerical model that calculated evapotranspiration and deep drainage for nine
combinations of soils and climates. The new curve was more robust than the other two. The
new curve was better able to produce a conductivity curve with a substantial drop during
the early stages of drying than the earlier improvement. It therefore generated smaller
amounts of more evenly distributed deep drainage compared to the spiked response to
rainfall produced by the earlier improvement.





## Introduction

The soil water retention function introduced by van Genuchten (1980) has been the

most popular parameterization (denoted VGN below; these and other abbreviations are

listed in Appendix A) to describe the SWRC in numerical models for unsaturated flow for

the past few decades (e.g., Kroes et al., 2017; Šimůnek and Bradford, 2008; Šimůnek et al.,

2016):

$$\theta(h) = \theta_r + (\theta_s - \theta_r)(1 + |\alpha h|^n)^{\frac{1}{n}-1}, \quad h \leq 0 \tag{1}$$

where $h$ denotes the matric potential in equivalent water column [L]. The volumetric water

content is denoted by $\theta$, with the subscript 's' denoting its value at saturation and the

subscript 'r' its residual, or irreducible, value. Parameters $\alpha$ [L$^{-1}$] and $n$ are shape

parameters.

Van Genuchten (1980) combined Eq. (1) with Mualem's (1976) conductivity model

and derived an analytical expression for the unsaturated hydraulic conductivity curve:

$$K(h) = K_s \frac{\left[1 - |\alpha h|^{n-1}(1 + |\alpha h|^n)^{\frac{1}{n}-1}\right]^2}{(1 + |\alpha h|^n)^{\frac{1}{2}-\frac{1}{2n}}} \tag{2}$$

where $K$ [LT$^{-1}$] is the soil hydraulic conductivity and $K_s$ [LT$^{-1}$] its value at saturation.

Hysteretic (Kool and Parker, 1987) and multimodal versions (Durner, 1994) of Eq.

(1) are available. Apart from the convenience of having analytical expressions for the

retention as well as the conductivity curve, the function's popularity derives from its



continuous derivative and its inflection point, which gives it considerable flexibility in
fitting observations.

Fuentes et al. (1991) warned that the asymptotic residual water content at the dry

end could lead to a non–converging integral of the retention curve, and showed how this
would mathematically lead to a physically impossible unlimited water uptake capacity of a
finite soil column. From their analysis follows that this can only be prevented if $n > 2$ in Eq.
(1), a condition which is often not satisfied.

Near saturation, the slope $\mathrm{d}\theta/\mathrm{d}h$ is not zero at zero matric potential. This implies

that the soil has pores that have at least one infinite principal radius according to the
Laplace–Young Law (Hillel, 1998, p. 46), which is physically unacceptable (see also Iden et
al., 2015). Durner (1994) noted this could lead to an infinite slope in the hydraulic
conductivity function of Mualem (1976) when the matric potential approached zero, and
Ippisch et al. (2006) showed that if $n < 2$ this would indeed be the case. The more recent
sigmoid curve of Fredlund and Xing (1994) and its modification by Wang et al. (2016), used
by Wang et al. (2018) and Rudiyanto et al. (2020) have the same problem (see Appendix B
for the proof). The curve of Assouline et al. (1998) is based on the Weibull distribution, and
therefore has a non–zero slope at zero matric potential when its fitting parameter $\eta$ is
smaller than 2, which was the case for 75% of the soils for which it was fitted. None of these
curves therefore offers a remedy to the problem associated with VGN.

Corrections for the conductivity curve were proposed by Vogel et al. (2001), Schaap

and van Genuchten (2006), and Iden et al. (2015), but these leave the effect of the non–
physical, very large pores on the SWRC intact and create an inconsistency between the
retention model and the conductivity model. For instance, Iden et al. (2015) clipped the





integral in the conductivity function at a matric potential $h_c$ somewhat below zero. In the
range between $h_c$ and zero, their modified unsaturated hydraulic conductivity increased
linearly with the water content (Iden et al. (2015), Fig. 1), which is not physically realistic
because the pore sizes that are being filled are increasing in size according to Eq. (1) or its
multimodal version (Peters et al., 2011). Only Ippisch et al. (2006) addressed the
underlying problem in the SWRC by introducing a non–zero air–entry value, thereby
eliminating excessively large pores whilst maintaining the mathematical consistency
between the expressions for the retention and the conductivity curves. In doing so they
sacrificed the continuity of the derivative of the VGN curve. Iden et al. (2015) suspected this
would pose a challenge to numerical solves of Richards' equation, but Ippisch et al.'s (2006)
numerical simulations ran without difficulty. Their equation scaled the sigmoid curve by its
value at the air–entry value $h_{ae}$ [L] and introduced a saturated section for $h > h_{ae}$.

$$\theta(h) = \begin{cases} \theta_r + (\theta_s - \theta_r)\left(\frac{1+|\alpha h|^n}{1+|\alpha h_{ae}|^n}\right)^{\frac{1}{n}-1}, & h \leq h_{ae} \\ \theta_s, & h > h_{ae} \end{cases}$$     (3)

This function is denoted VGA below.

The smooth, sigmoidal shape of VGN resembles many observed curves for which the

data points in the wet range were obtained by equilibrating cylindrical soil samples at well–
defined matric potentials and determining the corresponding water content by weighing
the sample (Klute, 1986, p. 644–647). Liu and Dane (1995) took into account the vertical
variation of the water content in such samples and demonstrated that a power–law SWRC
with a well–defined air–entry matric value but without inflection point can produce a



sigmoid–type apparent SWRC if the non–uniform distribution of water in the sample is
ignored. A series of data points suggesting a smooth SWRC therefore does not intrinsically
contradict the existence of a discrete non–zero air–entry value, corroborating the
correction to VGN by Ippisch et al. (2006).

In a separate development, several researchers argued that in the dry range, water is

bound to the soil by adsorptive rather than capillary forces. Usually, a logarithmic term that
allowed the adsorbed water content to go to zero at a prescribed matric potential was
added to a capillary term. The former would dominate in the dry range and become
negligible as the soil became wetter (e.g., Campbell and Shiozawa, 1992; Fayer and
Simmons, 1995; Khlosi et al., 2006; Peters, 2013). The logarithmic relationship was based
on the sorption theory of Bradley (1936). It removed the asymptote and the associated
problem of the non–converging integral of the SWRC that Fuentes et al. (1991) warned
about. Rossi and Nimmo (1994) presented a junction model in which a critical matric
potential separated purely capillary bound water (described by a variation of the Brooks–
Corey (1964) model) from solely adsorbed water. By doing so they avoided the problem of
many of the other models that would have some capillary bound water still present in the
soil below the matric potential at which the adsorbed water content had gone to zero.

Madi et al. (2018) generalized the analysis of Ippisch et al. (2006) and applied it to

18 parameterizations of the SWRC to verify that the slope of the hydraulic conductivity near
saturation would remain finite. Apart from Eq. (3), only the expressions developed by
Brooks and Corey (1964) (denoted BCO), Fayer and Simmons (1995) (denoted FSB), and
the junction model of Rossi and Nimmo (1994) (denoted RNA) satisfied this requirement.
In the latter case, the equation had to be modified by removing a modification that



smoothed the curve near saturation. All these equations have a power law relationship
between the water content and the matric potential, and therefore do not have the sigmoid
shape of VGN and VGA.

As noted above, the introduction of a non–zero air–entry value by Ippisch et al.

(2006) eliminated the unphysically large slopes of the hydraulic conductivity according to
Mualem (1976). The approach of Rossi and Nimmo (1994) resolved the issue of the
asymptotic behavior in the dry range. The objective of this paper therefore is to combine
Rossi and Nimmo's (1994) model for the dry range with the VGA model of Ippisch et al.
(2006) to arrive at a SWRC (denoted RIA) that has a non–zero air–entry value, a sigmoid
shape in the intermediate range, a dry branch that can reach zero water content at a finite
matric potential, and therefore a finite integral. We will also develop an explicit expression
for the unsaturated hydraulic conductivity based on this SWRC. For completeness, a
generalized expression for multimodal SWRCs will also be derived.

Together with the other functions that lead to physically acceptable behavior of the

hydraulic conductivity near saturation (BCO, FSB, RNA, and VGA), RIA will be fitted to 21
soils selected from the UNSODA database (National Agricultural Library, 2017; Nemes et al.,
2001) that cover a wide range of textures (Madi et al., 2018). For comparison VGN is also
included, in view of its de facto status as the standard parameterization for the SWRC. All
three versions with the sigmoid shape (VGN, VGA, and RIA) will be tested in a simulation
study for different combinations of soil types and climates.

**Theory**
**The Soil Water Retention Curve**



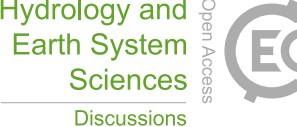

The junction model of Rossi and Nimmo (1994) has a SWRC with a logarithmic dry

branch without residual water content. The parameterization proposed by Ippisch et al.
(2006) combines the sigmoid shape of van Genuchten (1980) with a non–zero air–entry
value. By setting the $\theta_r$ in Ippisch et al. (2006) to zero, we can combine the two models to
give the following parameterization:

$$\theta(h) = \begin{cases} 0, & h \leq h_d \\ \theta_s \beta \ln\left(\frac{h_d}{h}\right), & h_d < h \leq h_j \\ \theta_s \left(\frac{1+|\alpha h|^n}{1+|\alpha h_{ae}|^n}\right)^{\frac{1}{n}-1}, & h_j < h \leq h_{ae} \\ \theta_s, & h > h_{ae} \end{cases} \tag{4}$$

where subscripts 'd' and 'ae' denote the value at which the water content reaches zero and
the air–entry value, respectively, and subscript 'j' indicates the value at which the
logarithmic and sigmoid branch are joined. Joining instead of adding the the logarithmic
and the sigmoid functions avoids potentially non-monotonic behavior (Peters et al., 2011).

The derivative of Eq. (4) is


$$\frac{d\theta}{dh}(h) = \begin{cases} 0, & h \leq h_d \\ \frac{\theta_s \beta}{h}, & h_d < h \leq h_j \\ \theta_s \alpha(n-1)|\alpha h|^{n-1}(1+|\alpha h_{ae}|^n)^{1-\frac{1}{n}}(1+|\alpha h|^n)^{\frac{1}{n}-2}, & h_j < h \leq h_{ae} \\ 0, & h > h_{ae} \end{cases} \tag{5}$$

Mass continuity dictates that the SWRC is continuous. At the air–entry value, this condition
is met irrespective of the parameter values. At $h_j$, continuity requires the following equality
to hold:






$$\beta \ln\left(\frac{h_d}{h_j}\right) = \left(\frac{1+|\alpha h_j|^n}{1+|\alpha h_{ae}|^n}\right)^{\frac{1}{n}-1} \tag{6}$$

In accordance with Rossi and Nimmo (1994) we also require the derivatives at $h_j$ to match,
leading to

$$\beta = (n-1)|\alpha h_j|^n (1+|\alpha h_{ae}|^n)^{1-\frac{1}{n}}(1+|\alpha h_j|^n)^{\frac{1}{n}-2} \tag{7}$$

Combining Eqs. (6) and (7) and solving for $h_d$ gives:

$$h_d = h_j \exp\left(\frac{1+|\alpha h_j|^{-n}}{n-1}\right) \tag{8}$$

This leaves $h_{ae}$, $h_j$, $\theta_s$, $\alpha$, and $n$ as fitting parameters.
The derivation of a multimodal curve analogous to that of Durner (1994) and
Zurmühl and Durner (1998) is straightforward if the values of $h_{ae}$ and $h_j$ are kept the same
for all contributing terms:
$$\theta(h) = \begin{cases} 0, & h \leq h_d \\ \theta_s \beta \ln\left(\frac{h_d}{h}\right), & h_d < h \leq h_j \\ \theta_s \sum_{i=1}^{k} w_i \left(\frac{1+|\alpha_i h|^{n_i}}{1+|\alpha_i h_{ae}|^{n_i}}\right)^{\frac{1}{n_i}-1}, & h_j < h \leq h_{ae} \\ \theta_s, & h > h_{ae} \end{cases} \tag{9}$$





where the modality is indicated by $k$. The weighting factors $w_i$ are bounded on the interval
[0,1] and their sum must equal 1 (Durner, 1994; Zurmühl and Durner, 1998). Requiring the
logarithmic and the multimodal branch as well as their derivatives to match at $h_\mathrm{j}$ leads to

$$\beta = \sum_{i=1}^{k} w_i (n_i - 1)\left|\alpha_i h_\mathrm{j}\right|^{n_i} (1 + |\alpha_i h_\mathrm{ae}|^{n_i})^{1-\frac{1}{n_i}} \left(1 + \left|\alpha_i h_\mathrm{j}\right|^{n_i}\right)^{\frac{1}{n_i}-2}$$  (10)

and

$$h_\mathrm{d} = h_\mathrm{j}\exp\left[\frac{\sum_{i=1}^{k} w_i (1+|\alpha_i h_\mathrm{ae}|^{n_i})^{1-\frac{1}{n_i}}\left(1+\left|\alpha_i h_\mathrm{j}\right|^{n_i}\right)^{\frac{1}{n_i}-1}}{\sum_{i=1}^{k} w_i (n_i-1)\left|\alpha_i h_\mathrm{j}\right|^{n_i}(1+|\alpha_i h_\mathrm{ae}|^{n_i})^{1-\frac{1}{n_i}}\left(1+\left|\alpha_i h_\mathrm{j}\right|^{n_i}\right)^{\frac{1}{n_i}-2}}\right]$$  (11)

The fitting parameters are $h_\mathrm{ae}$, $h_\mathrm{j}$, $\theta_\mathrm{s}$, $\alpha_i$, $n_i$, and $w_i$.

### The Unsaturated Hydraulic Conductivity Curve

Kosugi (1999) proposed the following conductivity model (see also Ippisch et al.,

2006):


$$K(h) = K\big(h(S_\mathrm{e})\big) = \begin{cases} K_\mathrm{s} S_\mathrm{e}^{\tau}\left(\dfrac{\int_{-\infty}^{h(S_\mathrm{e})}|h|^{-\kappa}\frac{\mathrm{d}S}{\mathrm{d}h}\mathrm{d}h}{\int_{-\infty}^{h_\mathrm{ae}}|h|^{-\kappa}\frac{\mathrm{d}S}{\mathrm{d}h}\mathrm{d}h}\right)^{\gamma} , & h < h_\mathrm{ae} \\ K_\mathrm{s}, & h \geq h_\mathrm{ae} \end{cases}$$  (12)

with $\gamma$, $\kappa$, and $\tau$ denoting shape parameters, $S_\mathrm{e}$ denoting the degree of saturation, and $S$
representing a variable running over all values of the degree of saturation from zero to its





actual value $S_e$. Mualem's (1976) conductivity model is a special case of Eq. (11), with $\gamma = 2$,
$\kappa = 1$, and $\tau = 0.5$.
The integrals in Eq. (12) that arise when Eq. (5) is used to find $\mathrm{d}S/\mathrm{d}h$ can be
evaluated analytically if $\kappa = 1$. The resulting conductivity function is

$$
K(h) = \begin{cases}
0, & h \leq h_\mathrm{d} \\[4mm]
K_\mathrm{s}\left(\beta\ln\left|\dfrac{h_\mathrm{d}}{h}\right|\right)^\tau \left[\dfrac{\dfrac{\beta}{|h_\mathrm{d}|} - \dfrac{\beta}{|h|}}{\dfrac{\beta}{|h_\mathrm{d}|} - \dfrac{\beta}{|h_\mathrm{j}|} + \dfrac{|\alpha h_\mathrm{ae}|^n}{|h_\mathrm{ae}|} - (1+|\alpha h_\mathrm{ae}|^n)^{1-\frac{1}{n}}F(h_\mathrm{j})}\right]^\gamma, & h_\mathrm{d} < h \leq h_\mathrm{j} \\[10mm]
K_\mathrm{s}\left(\dfrac{1+|\alpha h|^n}{1+|\alpha h_\mathrm{ae}|^n}\right)^{\frac{\tau}{n}-\tau} \left\{\dfrac{\dfrac{\beta}{|h_\mathrm{d}|} - \dfrac{\beta}{|h_\mathrm{j}|} + (1+|\alpha h_\mathrm{ae}|^n)^{1-\frac{1}{n}}[F(h) - F(h_\mathrm{j})]}{\dfrac{\beta}{|h_\mathrm{d}|} - \dfrac{\beta}{|h_\mathrm{j}|} + \dfrac{|\alpha h_\mathrm{ae}|^n}{|h_\mathrm{ae}|} - (1+|\alpha h_\mathrm{ae}|^n)^{1-\frac{1}{n}}F(h_\mathrm{j})}\right\}^\gamma, \\[2mm]
& h_\mathrm{j} < h \leq h_\mathrm{ae} \\[4mm]
K_\mathrm{s}, & h > h_\mathrm{ae}
\end{cases}
$$

(13a)

where

$F(h) = \dfrac{|\alpha h|^n(1+|\alpha h|^n)^{\frac{1}{n}-1}}{|h|}$  (13b)

Other conductivity models are available and can be selected if desired, for instance using
the framework presented by Weber et al. (2019).





The conductivity function associated with the multimodal soil water retention
function cannot be expressed in analytical form. For that case, the degree of saturation for
any $h$ can be found with Eq. (9) and the corresponding hydraulic conductivity determined
with Eq. (12) or another conductivity model.

## Materials and Methods

We selected 21 soils from the UNSODA database that had sufficient retention data
and together covered the textures represented in UNSODA. We then fitted Eq. (4) to these
soils using a Shuffled Complex Evolution algorithm (see Madi et al. (2018) and Appendix C
for details of the algorithm and the fitting procedure). We slightly modified the fitting code
used by Madi et al. (2018) to generate output that can more readily be converted to the
MATER.IN input file for Hydrus–1D, the numerical model used in this study. We therefore
refitted BCO, FSB, RNA, VGN, and VGA as well.
One–dimensional simulations for all combinations of three soils and three climates
were carried out to examine how the choice of parameterization affected fluxes at the soil
surface and in the subsoil. The model Hydrus–1D version 4.16.0110 (Šimůnek et al., 2013,
PC-progress website) was used for this purpose. The selected soils were a loamy sand
(UNSODA identifier: 2104), a silty loam (3261), and a clay (1181). Weather records were
generated from climate parameters based on weather data from Colombo (Sri Lanka,
monsoon climate), Tamale (Ghana, semi–arid climate), and Ukkel (Belgium, temperate
climate). Table 1 gives the most relevant statistics of the weather records. In order to
highlight the effects of the air–entry value and the logarithmic dry end of the SWRC, we
used the sigmoidal VGN, VGA, and RIA parameterizations in the simulations. The



Supplement details the generation of the weather records, the set–up of the simulations,
and the simulation results.

Table 1. Average cumulative monthly and annual rainfall and potential evapotranspiration
of the three test climates calculated from 1000–year generated weather records.

|  | Monsoon | | Semi–arid | | Temperate | |
|---|---|---|---|---|---|---|
| Month | Rain (mm) | $ET_{pot}$ (mm) | Rain (mm) | $ET_{pot}$ (mm) | Rain (mm) | $ET_{pot}$ (mm) |
| Jan | 94.0 | 160.5 | 14.3 | 159.4 | 69.0 | 12.7 |
| Feb | 83.3 | 156.0 | 12.9 | 158.1 | 55.6 | 18.8 |
| Mar | 217.6 | 166.8 | 13.7 | 185.0 | 59.4 | 34.4 |
| Apr | 233.8 | 160.0 | 34.1 | 177.7 | 57.1 | 51.1 |
| May | 237.4 | 164.0 | 117.6 | 166.6 | 24.5 | 73.9 |
| Jun | 138.5 | 164.7 | 140.9 | 151.4 | 24.4 | 83.3 |
| Jul | 139.4 | 172.4 | 141.8 | 153.2 | 23.2 | 86.3 |
| Aug | 137.1 | 174.4 | 213.5 | 145.0 | 20.9 | 73.7 |
| Sep | 133.0 | 164.7 | 214.7 | 136.9 | 23.8 | 50.6 |
| Oct | 325.8 | 142.3 | 76.6 | 154.1 | 38.0 | 30.8 |
| Nov | 328.8 | 128.1 | 12.9 | 149.3 | 49.1 | 16.1 |
| Dec | 112.0 | 150.9 | 12.9 | 151.9 | 51.2 | 11.2 |
| Annual sum | 2180.5 | 1904.8 | 1005.8 | 1888.7 | 496.1 | 542.9 |



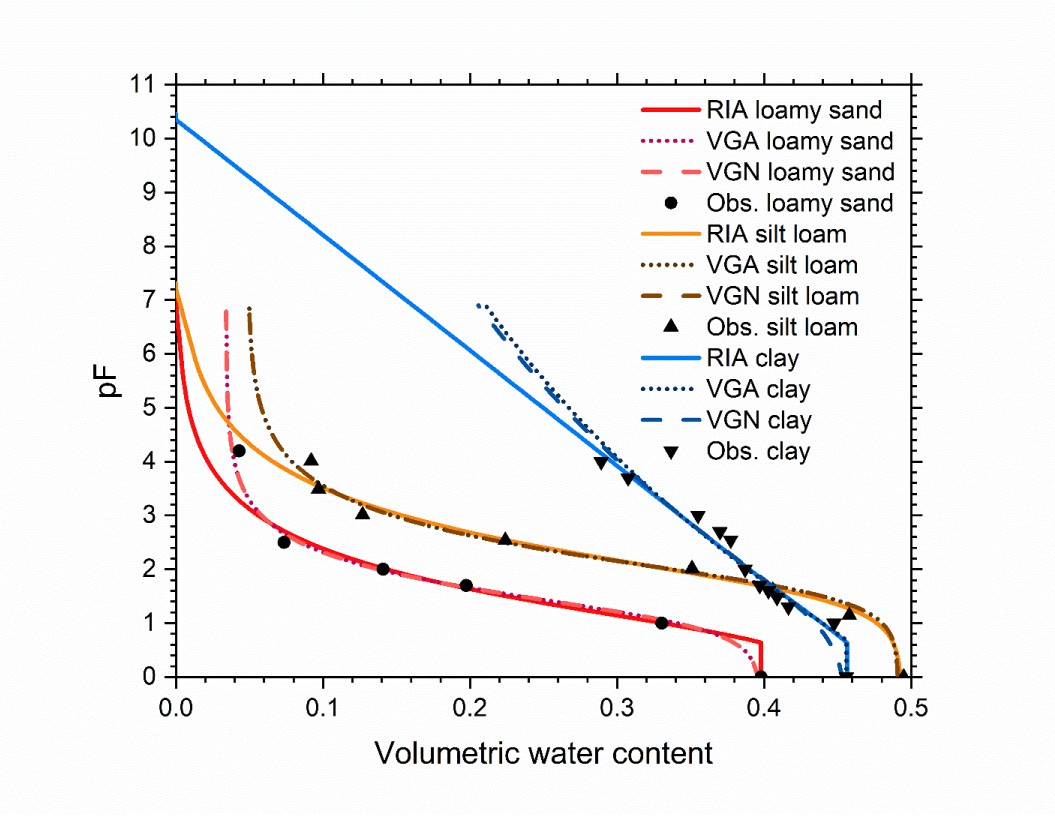


Figure 1. Soil water retention data of the soils used in the numerical simulations, and the

retention curves fitted to these data for three parameterizations with a sigmoid midsection

of the curve: the original model by van Genuchten (1980) (VGN), the modification thereof

by Ippisch et al. (2006) (VGA), and the further modification introduced in this paper (RIA).

The different parameterizations of the SWRCs (Table 2, Fig. 1) were used to generate

tables of the soil water retention and conductivity curves that were provided as input to

Hydrus–1D. Mualem's (1976) model for the unsaturated soil hydraulic conductivity was

used to generate relative hydraulic conductivities (scaled by $K_s$). These were converted to





unscaled conductivities (Fig. 2) by multiplying with the $K_s$–value according to the UNSODA
database, and then included in the tables.

Table 2. Parameters of the fit to data of the three parameterizations of the soil water
retention curve used for the numerical simulations.

| Soil | Parameter | $\theta_r$ | $\theta_s$ | $\alpha$ (cm$^{-1}$) | $n$ | $h_{ae}$ (cm) | $h_j$ (cm) |
|------|-----------|------------|------------|------------|-----|----------------|-------------|
| Clay | RIA | – | 0.45666 | 0.70200 | 1.0543 | −4.2523 | −205.65 |
| | VGA | $2.12\times10^{-6}$ | 0.45603 | 1.7047 | 1.0543 | −4.8324 | – |
| | VGN | $3.34\times10^{-6}$ | 0.45616 | 0.14265 | 1.0571 | – | – |
| Silt | RIA | – | 0.49346 | 0.023340 | 1.3691 | $-2.361\times10^{-3}$ | $-1.0739\times10^{6}$ |
| loam | VGA | 0.048871 | 0.49122 | 0.018365 | 1.5158 | $-2.081\times10^{-3}$ | – |
| | VGN | 0.048816 | 0.49134 | 0.018425 | 1.5149 | – | – |
| Loamy | RIA | – | 0.39801 | 0.17096 | 1.4106 | −4.3581 | $-7.7464\times10^{5}$ |
| sand | VGA | 0.034209 | 0.39771 | 0.069661 | 1.6395 | −0.016234 | – |
| | VGN | 0.034133 | 0.39772 | 0.069707 | 1.6389 | – | – |



For the clay soil, the recorded value (178 cm d$^{-1}$) was such that it can be assumed
that macropore flow contributed to its value. We therefore also ran simulations with a $K_s$ –
value of 1.25 cm d$^{-1}$. This value was adopted from soil 1182 from the UNSODA database,
from the same location.



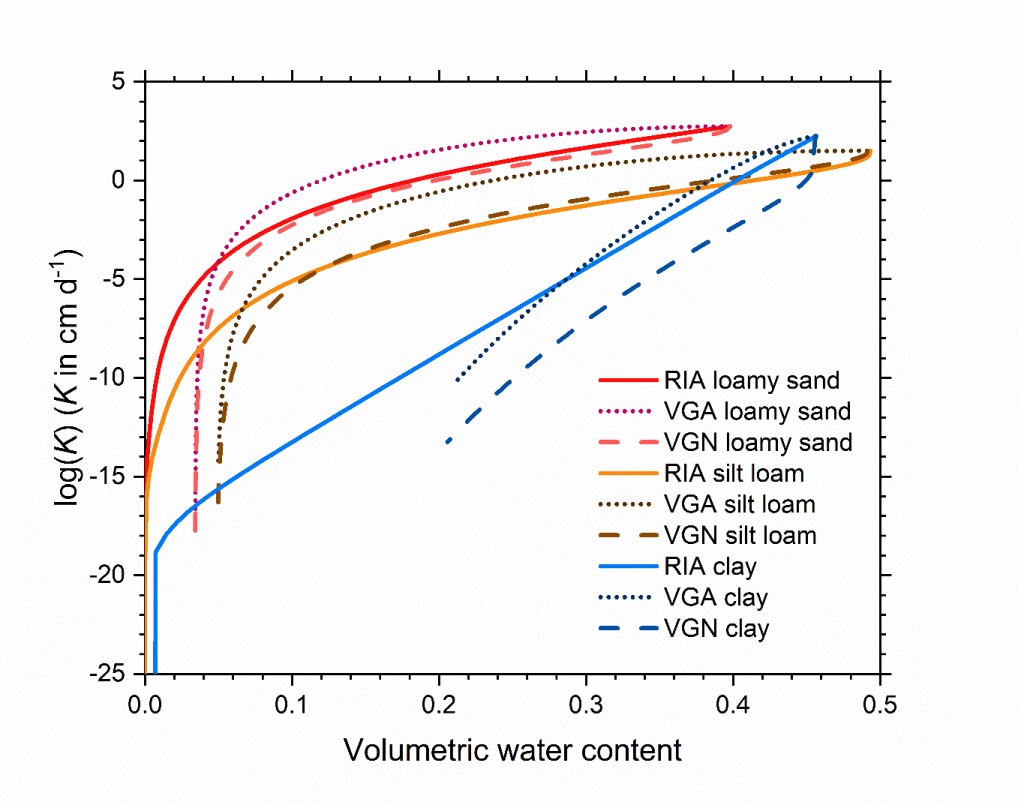


Figure 2. The unsaturated soil hydraulic conductivity curves derived from the fitted
retention curves depicted in Fig.1 and scaled by soil–specific values of the hydraulic
conductivity at saturation.

## Results and Discussion

### Fitted Curves for 21 Soils

The fitted parameter values (Tables C1–C4) and the associated curves (Figs. C2–C5)

are presented in Appendix C. The extra parameter of RIA compared to RNA and FSB gives it
a clear advantage in fitting the full water content range. The sigmoid shape of RIA provides
a better fit near the air entry value while still providing good fits of the drier data points





(e.g. 1121 in Fig. C3, all soils in Figs. C4–C5). In the wet range, the sigmoid curves (VGN,
VGA, RIA) outperform the power–law curves (BCO, FSB, RNA). In almost all cases, VGA and
RIA look very similar. Figure C3 shows multimodality in the data for five out of six soils that
cannot be reproduced by any of the parameterizations.

Many of the UNSODA soils have one retention data point at saturation and the next

at $h = -10$ cm. The air–entry value of many sandy soils is within that range. The fitting
routine struggles to fit $h_{ae}$ to these data for VGA and RIA because the sigmoid shape of the
van Genuchten parameterization has such flexibility that it can fit the intermediate range
well for a range of $h_{ae}$–values. We therefore recommend making $\theta(h)$–measurements at one
or two matric potentials between zero and $-10$ cm for coarse–textured soils.

Eleven of the 21 soils have residual water contents for VGN and/or VGA that exceed

0.05 (up to 0.263), mostly in loamy sands, loams, and clays (Tables C1–C4). All of these and
several others have dry–end data points with water contents that appear too high (Figs. C2–
C5). These water contents may have been overestimated due to the lack of equilibrium
reported by Bitelli and Flury (2009). The parameterizations without asymptote (FSB, RNA,
RIA) generally have more plausible fits based on visual inspection of the graphs, but
because they do not follow the upward tail of the data in the dry end, their RMSE values for
the cases with suspicious dry–end data points are larger than those of VGN and VGA (Tables
C5–C8).

Especially for fine–textured soils, the lack of data points for air dryness or oven

dryness (Figs. C3–C5) causes the fitting procedure to treat $\theta_r$ for the asymptotic dry
branches and $h_d$ for the logarithmic dry branches as pure fitting parameters for the drier
range of the data. This range exceeds pF4.2 in only one case and in several cases likely





suffers from lack of equilibrium (Bitelli and Flury, 2009). This leads to unrealistically high
values for both in many cases. If applications are envisioned for which low water contents
are expected, it would be better to have some data in the dry range and ensure equilibrium
has been achieved before fitting RIA. One could even add a virtual data point with zero
water content at pF6.8, since this is the value that is approximated by many laboratory
ovens (Schneider and Goss, 2012). We did not do so here to be able to observe how the
various parameterizations perform on frequently reported data ranges.

Figure C5 showcases a peculiarity of FSB. The original version by Fayer and

Simmons (1995) allowed capillary bound water to be present even if the adsorbed water
content was down to zero. We adopted the correction by Madi et al. (2018, Eq. (S12a)) that
forced the amount of water bound by capillary forces to zero if the adsorbed water content
goes reaches zero at pF6.8. In the case of clay soils, the parameter values are such that a
substantial amount of capillary bound water is still present at pF6.8, leading to a sudden
cut–off at pF6.8.

The hydraulic conductivity curves based on water retention curve parameters and

$K_s$ very poorly match the data in most cases (Fig. C6–C9). We note here that all but one (soil
4450, Fig. C8) sets of unsaturated conductivity data were obtained in the field, while all
retention data were laboratory data on drying samples. The reported $K_s$–values were
probably measured in a separate experiment (possibly in the laboratory), in which case a
mismatch between $K_s$ and the unsaturated conductivity data is to be expected. The poor
match with field data notwithstanding, the graphs can be used to study the effect of the
parameterization on the shape of the $K(\theta)$ curve. The comparison between measured and
modeled shapes of the conductivity curves is inconclusive.



In many soils, regardless of texture, RIA's $K(\theta)$ curve drops of much slower in the
dry range than those of VGN and VGA (Fig. C6–C9), a consequence of the ability of the
underlying SWRC to reach zero water content at a finite matric potential. Near saturation,
RIA's $K(\theta)$ curve often drops off sharply before leveling off, in stark contrast to that of VGA,
which remains high in the wet range. Given the similarity in the $\theta(h)$ curves of VGA and RIA,
the difference in their $K(\theta)$ curves is remarkable. RIA's $K(\theta)$ curve is the only one that can
drop off sharply near saturation, level out somewhat in the mid-range and drop ever more
sharply in the dry range. It is below many of the other curves in the wet and mid-range, and
above most of them in the dry range.
Peters et al. (2011) developed parameter constraints to ensure physically plausible
shapes of the SRWC and the conductivity curve. For FSB, the criterion for non-monotonicity
of the conductivity curve is not met for soil 4450 (Fig. C8), resulting in $K$ increasing with
decreasing water content near saturation.

**Model Simulations**
In total, 21 out of 36 combinations of soil–climate–parameterization ran to
completion. Runoff did not occur in any of the successful model runs. Convergence was not
achieved for any of the runs for the clay soil with the reduced $K_s$–value. For the clay soil
with the high $K_s$–value, only RIA ran to completion. The discontinuity of the first derivative
at the air-entry value did not cause numerical problems. In fact, the replacement of any
parametric expression by a look-up table creates a discontinuity of the first derivative at
every point of the look-up table.



Table 3 lists the main mean annual fluxes calculated from the simulation study. The
median flux was produced by RIA for all combinations of soil and climate. The mean annual
actual transpiration between the three parameterizations differed by more than 10% from
the median only for the loamy sand under a temperate climate.  Actual evaporation never
deviated more than 10%. The amount of water leaving the soil profile differed substantially
between parameterizations for the semi-arid and temperature, especially for VGA (20–46%
deviation from the median).
The daily data revealed significant differences on smaller time scales that will be
relevant if reactive solute transport is of interest (see the Supplement). The fluxes
generated by the VGA parameterizations responded more quickly and strongly to the
rainfall signal, with VGN and RIA giving a more delayed and smooth response. The SWRCs
(Fig. 1) offer no explanation for this, but Fig. 2 shows that VGA's hydraulic conductivity in
the wet and intermediate water content range for all three soils is considerably higher than
that of VGN and RIA, except for clay, where it is only moderately higher than RIA's and
drops below RIA at a water content of 0.28. Thus, small differences between SWRCs can
have a significant influence on soil water flow simulations through their effect on the soil
hydraulic conductivity curve, an effect that the reviews by Leij et al. (1997) and Assouline
and Or (2013) took into consideration to some degree, but was not considered in several
other studies that compared different parameterizations (e.g., Rossi and Nimmo, 1994;
Assouline et al., 1998; Cornelis et al., 2005; Khlosi et al., 2008).





Table 3. Average of the annual sums of the actual transpiration and evaporation, and of the
outflow across the lower boundary of the simulated soil profiles. The averages were
calculated for the final six years of the simulation periods. The values in parentheses are
scaled with respect to the corresponding value for RIA.

| Climate | Soil | Actual transpiration (mm) | | | Actual evaporation (mm) | | | Downward flux at 2 m depth (mm) | | |
|---|---|---|---|---|---|---|---|---|---|---|
| | | RIA | VGA | VGN | RIA | VGA | VGN | RIA | VGA | VGN |
| Monsoon | clay | 962 | – | – | 543 | – | – | 876 | – | – |
| | silt loam | 1114 | 1044 (0.94) | 1112 (1.06) | 541 | 554 (1.02) | 534 (0.96) | 711 | 763 (1.07) | 718 (0.94) |
| | loamy sand | 1037 | 976 (0.94) | 1024 (1.05) | 480 | 440 (0.92) | 461 (1.05) | 863 | 959 (1.11) | 896 (0.93) |
| Semi–arid | clay | 585 | – | – | 319 | – | – | 294 | – | – |
| | silt loam | 657 | 588 (0.90) | 653 (1.11) | 297 | 303 (1.02) | 287 (0.95) | 164 | 226 (1.37) | 177 (0.78) |
| | loamy sand | 585 | 540 (0.92) | 572 (1.06) | 248 | 222 (0.89) | 234 (1.06) | 290 | 363 (1.25) | 319 (0.88) |
| Temperate | clay | 202 | – | – | 163 | – | – | 167 | – | – |
| | silt loam | 281 | 236 (0.84) | 271 (1.15) | 150 | 151 (1.01) | 143 (0.95) | 99 | 144 (1.46) | 114 (0.79) |
| | loamy sand | 224 | 205 (0.92) | 214 (1.04) | 130 | 118 (0.91) | 124 (1.05) | 174 | 208 (1.20) | 190 (0.91) |




## Summary and Conclusions


The improvements incorporated in the RIA parameterization for the first time

remove problems of the popular VGN model in both the wet and the dry range while
retaining the desirable sigmoid shape in the mid–range. This shape allows its multimodal
version to represent SWRCs with multiple humps. RIA offers a wider range of shapes for the
conductivity curve than any other parameterization that does not lead to the unphysical
behavior near saturation that was revealed by Durner (1994) and Ippisch et al. (2006) for
VGN and by Madi et al. (2018) for 14 other parameterizations. RIA also proved to be more
robust during numerical simulations than VGN itself as well as its modification VGA, which
still has a non–physical asymptote at a non–zero residual water content. The deep drainage
generated by RIA was more spread out and smaller than the spiked response to rainfall
produced by VGA, probably because RIA was better able to produce a conductivity curve
with a substantial drop during the early stages of drying. We therefore hope that RIA or its
multimodal version will be adopted for use in numerical simulations of soil water flow. The
catalogue of parameters for 21 soils in Appendix C may be of help for such simulations.



# Appendix A. List of Abbreviations

BC: Brooks and Corey (1964)

BCO: parameterization of the SWRC according to the original Brooks and Corey (1964)
equation

FSB: parameterization of the SWRC according to the BC–based version of Fayers and
Simmons (1995)

RIA: parameterization of the SWRC that combines RNA and VGA

RNA: parameterization of the SWRC according to the junction model of Rossi and Nimmo
(1994)

SWRC: soil water retention curve

RMSE: root mean square error

UNSODA: unsaturated soil hydraulic properties database

VGA: parameterization of the SWRC according to Ippisch et al. (2016)

VGN: parameterization of the SWRC according to the original van Genuchten (1980)
equation



## Appendix B. Assessing the near-saturation behavior of recently

## developed soil water retention and hydraulic conductivity curves

Madi et al. (2018) developed a criterion that needs to be met to avoid unphysical

behavior of the unsaturated soil hydraulic conductivity curve near saturation:

$$\lim_{h \uparrow 0} \left( |h|^{-\kappa} \frac{\mathrm{d}\theta}{\mathrm{d}h} \right) = 0 \tag{B1}$$

where $\kappa$ is a fitting parameter ($> 0$) that appears in a several parameterizations of the

unsaturated soil hydraulic conductivity curve based on the capillary bundle

conceptualization.

Fredlund and Xing (1994) introduced the following parameterization for the soil

water retention curve:

$$\theta(h) = \left[ \frac{-\ln\left(1 + \frac{|h|}{|h_r|}\right)}{\ln\left(1 + \frac{b}{|h_r|}\right)} + 1 \right] \frac{\theta_s}{\left\{ \ln\left[ e + \left(\frac{|h|}{a}\right)^n \right] \right\}^m} \tag{B2}$$

where the subscript 'r' denotes the value when the residual water content is reached, and $a$

[L], $n$, and $m$ are fitting parameters. The first term on the right-hand side is a correction

term that forces the water content to zero for $h = -b$, with $b$ equal to $10^7$ cm (Fredlund and

Xing, 1994) or $6.3 \times 10^6$ cm (Wang et al., 2016). Wang et al. (2016) also modified the

correction factor to give

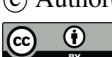


$$\theta(h) = \left[\frac{-\ln\left(1+\frac{c|h|}{|h_{\mathrm{r}}|}\right)}{\ln\left(1+\frac{bc}{|h_{\mathrm{r}}|}\right)} + 1\right]\frac{\theta_s}{\left\{\ln\left[\mathrm{e}+\left(\frac{|h|}{a}\right)^n\right]\right\}^m} \tag{B3}$$


where $c$ has to be small and positive. The derivative of Eq. (B3) is

$$\frac{\mathrm{d}\theta}{\mathrm{d}h} = \frac{-\theta_{\mathrm{s}}}{\left\{\ln\left[\mathrm{e}+\left(\frac{|h|}{a}\right)^n\right]\right\}^m}\left\{\frac{mn|h|^{n-1}\left[\ln\left(1+\frac{bc}{|h_{\mathrm{r}}|}\right)\ln\left(1+\frac{c|h|}{|h_{\mathrm{r}}|}\right)-1\right]}{a^n\left[\mathrm{e}+\left(\frac{|h|}{a}\right)^n\right]\ln\left[\mathrm{e}+\left(\frac{|h|}{a}\right)^n\right]} - \frac{\ln\left(1+\frac{bc}{|h_{\mathrm{r}}|}\right)}{\frac{|h_{\mathrm{r}}|}{c}+|h|}\right\} \tag{B4}$$


The derivative of Eq. (B2) follows by setting $c$ equal to 1 in Eq. (B4). Combining Eq.(B4)
with Eq. (B1) results in the following requirement:

$$\lim_{h\uparrow 0}\left[\frac{\theta_{\mathrm{s}}mn}{\mathrm{e}a^n}|h|^{-\kappa+n-1} + \frac{c\theta_{\mathrm{s}}\ln\left(1+\frac{bc}{|h_{\mathrm{r}}|}\right)}{|h_{\mathrm{r}}|}|h|^{-\kappa}\right]=0 \tag{B5}$$


The limit goes to zero if and only if the exponents of in both terms are positive. Hence, $\kappa <$
$n$–1 and $\kappa < 0$. The first requirement may be met for some soils, but the second violates
the physical constraint that $\kappa$ cannot be negative (Madi et al., 2018). Therefore, neither
Fredlund and Xing's (1994) nor Wang et al.'s (2016) parameterization lead to unsaturated
hydraulic conductivity curves that exhibit physically realistic behavior near saturation.
Wang et al. (2018) added a modification in the dry end of Wang et al. (2016), and
Rudiyanto et al. (2020) in turn used Wang et al.'s (2018) curves. Because the problem near
saturation was not resolved, these two hydraulic conductivity models suffer from the same
problem near saturation.





## Appendix C. Fitted Parameters and Root Mean Square Error for Six Parameterizations of the Soil Water Retention Curve applied to Data of 21 Soils

The UNSODA soils selected for parameter fitting are grouped in Tables C1–C4 according to their texture classification according to Twarakavi et al. (2010). Sand is a major constituent of the mineral soil in Tables C1 and C2, silt in Table C3, and clay in Table C4. Figure C1 shows the textural composition of the soils. For each soil, the parameter values for six parameterizations are given, resulting in a total of 126 SWRC parameterizations. The Root Mean Square Errors (RMSE) for all fits are listed in Tables C5–C8. The saturated hydraulic conductivities of the soils as given by the UNSODA database are given in Table C9.

Madi et al. (2018) provide an analysis of the underlying functions of all parameterizations tested here, except RIA. The meaning of all variables except $\lambda$ is explained in the main text. Variable $\lambda$ is the power to which the factor ($h$ / $h_{ae}$) is raised in the power–law segments of the SWRCs of BCO, FSB, and RNA. In RNA, $\lambda$ is expressed as a function of the fitting parameters, and therefore does not appear in the tables.

The SWRCs defined by these parameterizations together with the data points on which they are based are given in Figs. C2–C5. The hydraulic conductivity curves according to Mualem (1976) that can be derived from the parameterizations (Eqs. (13a) and (13b) for RIA, equations for the other parameterizations in Madi et al. (2018)) are plotted in Figs. C6–C9. We plotted $K$ as a function of $\theta$ because this relationship is less hysteretic than $K(h)$


(Koorevaar et al., 1983, p. 141) and some of the UNSODA data only provided $K(\theta)$ data
points. The conductivity data available in UNSODA are also plotted, but these were not used
to fit the curves to.

The comparison of theoretical conductivity curves based on water retention data

with the measured values showed a sometimes substantial deviation between hydraulic
conductivities measured at saturation and the $K_s$ value used to create the theoretical curves.
The latter value was obtained by a query that made the database return a $K_s$ value for the
soils that met all other criteria also included in the query. The conductivity data displayed
in the plots were separately obtained by requesting the database to return a report of
tabular data for each of the 21 soils. The reasons for the discrepancies between the queried
value and the tabulated may reflect separate experiments for measuring unsaturated and
saturated values of $K$.

There also are obvious differences between the water content at saturation between

the retention and the conductivity data. In all but one case (soil 4450, Fig. C8) the hydraulic
conductivity observations were made in the field whereas the retention data used were all
obtained from drying experiments in the laboratory. Scale differences, effects of air
enclosure and hysteresis in the field, and differences between different measurement
techniques explain these differences. We can therefore only compare the shape of the
theoretical curves with those of the data clouds.



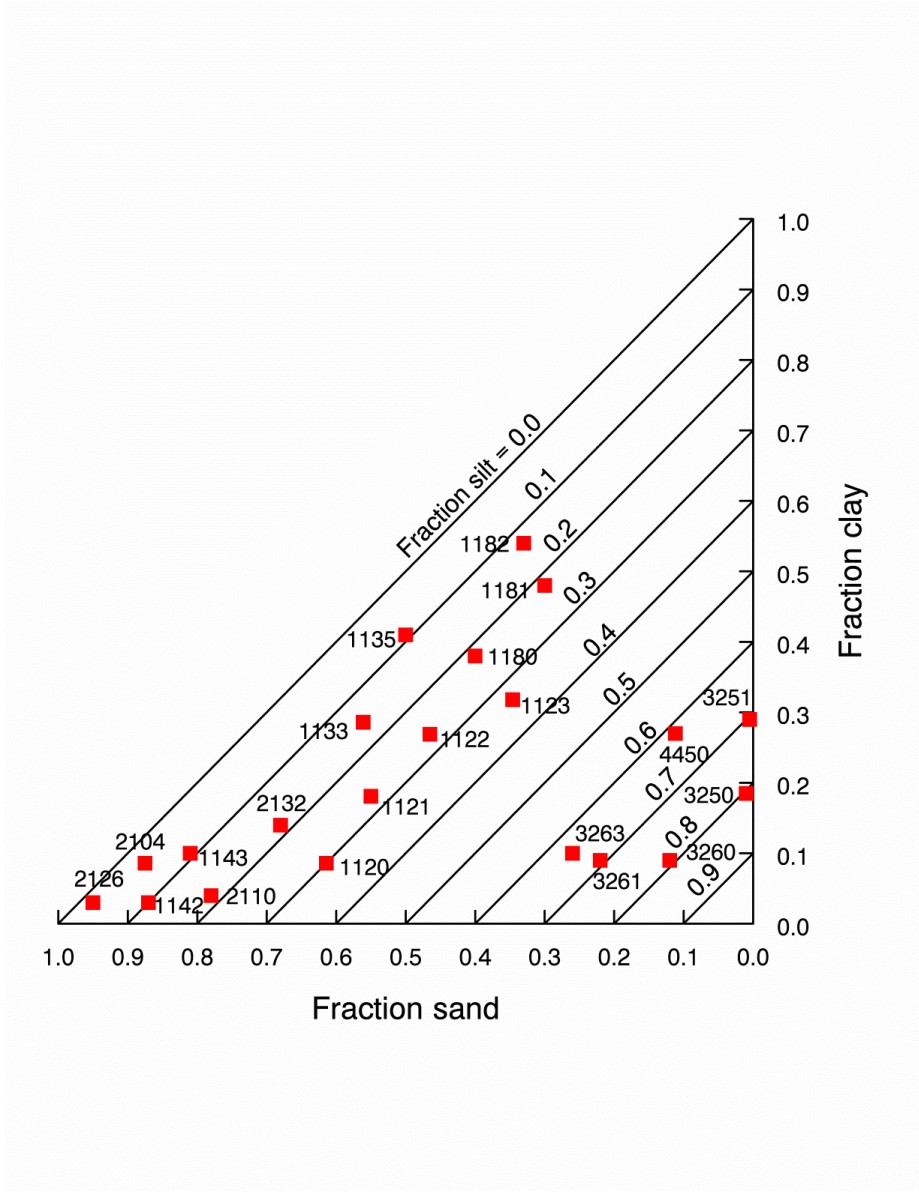


Figure C1. The positions in the texture triangle of the selected soils, indicated by their

UNSODA identifier. (Taken form Madi et al., 2018)






The RMSE was calculated from the weighted sum of squares of the differences
between calculated and observed water contents and pressure heads. The weights equaled
the estimated scaled standard deviations of the individual water retention observations
(pairs of matric potential and water content values). The standard deviations of the water
content observations were scaled to have an average value of 0.2. The scale factor needed to
arrive at this value was then applied to the standard deviations of the matric potential
values as well. This ensured that the weighting of water contents and matric potential
values was consistent with the original standard deviations of both. The scaling greatly
improved the efficiency of the parameter fitting procedure. The squared difference between
a single observed water content and its fitted value during a given iteration of the
parameter fitting algorithm, taking into account observation errors in both the water
content and the matric potential, is

$$\left(\frac{\theta_{\text{fit}}-\theta_{\text{obs}}}{\sigma_{\theta,\,\text{scaled}}+\sigma_{h,\text{scaled}}\frac{d\theta}{dh}\Big|_{h_{\text{obs}}}}\right)^{2}$$    (C1)

where $\sigma_{a,\,\text{scaled}}$ is the scaled standard deviation of the variable $a$, subscript 'fit' signifies a
fitted value of the subscripted variable, and subscript 'obs' a measured value (Madi et al.,
2018). The slope of the SWRC in the denominator is estimated from the fitted
parameterization using the parameter values that have been fitted in the iteration that is
currently being tested.

The scale factors applied to the observation error standard deviations varied

between soils but not between parameterizations fitted to the same soil. RMSE values of





different parameterizations valid for a particular soil can therefore be readily compared.
Comparisons between soils only give a rough indication. When the water contents at which
measurements were taken differ strongly from one soil to another, the comparison of their
RMSE values is less reliable.

In case the measured water contents were obtained at hydrostatic equilibrium, the

fitted water content calculated directly from the matric potential could not be compared to
the observed water content. To approximate the soil sample on which the observation was
made, a hypothetical soil slab of the same height as the sample was divided into 20
horizontal layers. If UNSODA did not specify the sample height, it was assumed to be 5.0 cm.
The matric potential in the center of each layer was determined from the given matric
potential, which was assumed to apply to the center of the sample. The water content of the
soil slab was then calculated as the average water content of its 20 layers. This water
content was used to calculate the difference between the observed and the fitted water
content. Figure C10 shows a comparison of retention points calculated for three soils based
directly on the RIA parameterization and based on the same parameterization applied to a
hypothetical sample of 5.0 cm height at hydrostatic equilibrium with the nominal matric
potential valid at the sample center. Deviations are small, even for the loamy sand.





Table C1. The fitting parameters and their values for six parameterizations for sandy soils in the A1
and A2 classifications of Twarakavi et al. (2010) from the UNSODA database (National Agricultural
Library, 2017; Nemes et al., 2001). The three–character parameterization label is explained in the
main text.

| | | | Soil (UNSODA identifier and classification according to Twarakavi et al. (2010)) | | |
| | | | 2126 A1 | 1142 A2 | 2104 A2 |
|---|---|---|---|---|---|
| Parameterization | Parameter | Unit | | | |
| BCO | $\theta_r$ | – | 1.63E–2 | 4.92E–5 | 2.27E–2 |
| | $\theta_s$ | – | 0.377 | 0.250 | 0.398 |
| | $h_{ae}$ | cm | −6.78 | −7.00 | −6.79 |
| | $\lambda$ | – | 0.846 | 0.210 | 0.434 |
| FSB | $\theta_s$ | – | 0.377 | 0.250 | 0.398 |
| | $\theta_a$ | – | 2.58E–2 | 6.26E–5 | 5.46E–2 |
| | $h_{ae}$ | cm | −6.76 | −7.00 | −6.73 |
| | $\lambda$ | – | 0.861 | 0.211 | 0.468 |
| RNA | $\theta_s$ | – | 0.378 | 0.250 | 0.398 |
| | $h_{ae}$ | cm | −6.37 | −7.00 | −6.17 |
| | $h_j$ | cm | −8.87E4 | −9.24E4 | −6.45E4 |
| | $h_d$ | cm | −3.60E5 | −1.07E7 | −9.73E5 |
| VGN | $\theta_r$ | – | 3.39E–2 | 9.42E–2 | 3.41E–2 |
| | $\theta_s$ | – | 0.376 | 0.242 | 0.398 |
| | $\alpha$ | cm$^{-1}$ | 6.85E–2 | 1.99E–2 | 6.97E–2 |
| | $n$ | – | 2.73 | 2.93 | 1.64 |
| VGA | $\theta_r$ | – | 3.39E–2 | 9.64E–2 | 3.42E–2 |
| | $\theta_s$ | – | 0.376 | 0.242 | 0.398 |
| | $\alpha$ | cm$^{-1}$ | 6.84E–2 | 1.98E–2 | 6.97E–2 |
| | $n$ | – | 2.73 | 3.05 | 1.64 |
| | $h_{ae}$ | cm | −0.101 | −0.240 | −1.62E–2 |
| RIA | $\theta_s$ | – | 0.378 | 0.245 | 0.398 |
| | $\alpha$ | cm$^{-1}$ | 0.239 | 2.68E–2 | 0.171 |
| | $n$ | – | 1.77 | 1.47 | 1.41 |
| | $h_{ae}$ | cm | −5.82 | −7.00 | −4.36 |
| | $h_j$ | cm | −1.76E6 | −4.35E5 | −7.75E5 |






Table C2. The fitting parameters and their values for six parameterizations for sandy soils in the A3
and A4 classifications of Twarakavi et al. (2010) from the UNSODA database (National Agricultural
Library, 2017; Nemes et al., 2001). The three–character parameterization label is explained in the
main text.

| Parameterization | Parameter | Unit | Soil (UNSODA identifier and classification according to Twarakavi et al. (2010)) | | | | | |
|---|---|---|---|---|---|---|---|---|
| | | | 1120 A3 | 1143 A3 | 2110 A3 | 2132 A3 | 1121 A4 | 1133 A4 |
| BCO | $\theta_r$ | – | 6.15E–6 | 2.71E–5 | 0.103 | 4.10E–5 | 2.64E–5 | 5.37E–5 |
| | $\theta_s$ | – | 0.311 | 0.279 | 0.348 | 0.303 | 0.350 | 0.330 |
| | $h_{ae}$ | cm | −10.0 | −7.00 | −25.5 | −8.00 | −10.0 | −206 |
| | $\lambda$ | – | 0.204 | 0.169 | 0.537 | 0.107 | 0.118 | 0.103 |
| FSB | $\theta_s$ | – | 0.311 | 0.279 | 0.348 | 0.308 | 0.346 | 0.330 |
| | $\theta_a$ | – | 4.33E–5 | 4.26E–4 | 0.213 | 0.298 | 0.324 | 0.310 |
| | $h_{ae}$ | cm | −10.0 | −7.00 | −25.7 | −3.24 | −10.0 | −206 |
| | $\lambda$ | – | 0.204 | 0.169 | 0.763 | 0.422 | 0.377 | 0.213 |
| RNA | $\theta_s$ | – | 0.311 | 0.279 | 0.351 | 0.303 | 0.352 | 0.330 |
| | $h_{ae}$ | cm | −10.0 | −7.00 | −20.3 | −8.00 | −10.0 | −220 |
| | $h_j$ | cm | −7.32E4 | −8.46E4 | −6.44E4 | −7.18E4 | −8.73E4 | −6.77E4 |
| | $h_d$ | cm | −9.95E6 | −3.18E7 | −2.35E6 | −8.06E8 | −3.88E8 | −8.68E8 |
| VGN | $\theta_r$ | – | 7.22E–2 | 9.24E–2 | 0.126 | 1.25E–4 | 1.70E–5 | 0.202 |
| | $\theta_s$ | – | 0.305 | 0.278 | 0.360 | 0.305 | 0.339 | 0.324 |
| | $\alpha$ | cm$^{-1}$ | 1.72E–2 | 4.66E–2 | 2.63E–2 | 5.73E–2 | 7.21E–3 | 7.34E–4 |
| | $n$ | – | 1.69 | 1.49 | 1.84 | 1.14 | 1.26 | 3.02 |
| VGA | $\theta_r$ | – | 7.37E–2 | 0.112 | 0.106 | 6.61E–4 | 1.47E–5 | 0.202 |
| | $\theta_s$ | – | 0.303 | 0.276 | 0.348 | 0.306 | 0.339 | 0.324 |
| | $\alpha$ | cm$^{-1}$ | 1.72E–2 | 4.41E–2 | 0.230 | 6.06E–2 | 7.16E–3 | 7.35E–4 |
| | $n$ | – | 1.71 | 1.66 | 1.56 | 1.14 | 1.27 | 3.00 |
| | $h_{ae}$ | cm | −9.37 | −6.81 | −25.3 | −8.47E−4 | −2.73E−2 | −12.5 |
| RIA | $\theta_s$ | – | 0.308 | 0.280 | 0.360 | 0.306 | 0.339 | 0.328 |
| | $\alpha$ | cm$^{-1}$ | 3.01E–2 | 6.39E–2 | 4.18E–2 | 6.08E–2 | 7.13E–2 | 1.30E–3 |
| | $n$ | – | 1.29 | 1.23 | 1.33 | 1.14 | 1.27 | 1.20 |
| | $h_{ae}$ | cm | −1.24E−3 | −4.96E−4 | −7.85 | −7.44E−4 | −9.11E−4 | −220 |
| | $h_j$ | cm | −1.71E6 | −7.01E6 | −9.37E6 | −7.81E6 | −2.04E6 | −5.07E6 |





Table C3. The fitting parameters and their values for six parameterizations for silty soils from the
UNSODA database (National Agricultural Library, 2017; Nemes et al., 2001). The three–character
parameterization label is explained in the main text.

| Para–meteri–zation | Para–me–ter | Unit | Soil (UNSODA identifier and classification according to Twarakavi et al. (2010)) | | | | | |
|---|---|---|---|---|---|---|---|---|
| | | | 3260 B2 | 3261 B2 | 3263 B2 | 3250 B4 | 3251 B4 | 4450 B4 |
| BCO | $\theta_r$ | – | 2.85E–6 | 3.42E–6 | 3.85E–7 | 3.85E–6 | 1.94E–6 | 2.77E–5 |
| | $\theta_s$ | – | 0.470 | 0.499 | 0.460 | 0.540 | 0.500 | 0.380 |
| | $h_{ae}$ | cm | −28.6 | −13.5 | −28.8 | −30.5 | −18.2 | −4.80 |
| | $\lambda$ | – | 0.281 | 0.256 | 0.255 | 0.183 | 9.56E–2 | 9.51E–2 |
| FSB | $\theta_s$ | – | 0.470 | 0.499 | 0.460 | 0.540 | 0.500 | 0.380 |
| | $\theta_a$ | – | 1.33E–5 | 6.49E–5 | 1.33E–5 | 0.173 | 0.431 | 0.320 |
| | $h_{ae}$ | cm | −28.6 | −13.5 | −28.8 | −30.0 | −10.9 | −0.882 |
| | $\lambda$ | – | 0.281 | 0.256 | 0.255 | 0.241 | 0.197 | 0.196 |
| RNA | $\theta_s$ | – | 0.470 | 0.499 | 0.460 | 0.540 | 0.500 | 0.380 |
| | $h_{ae}$ | cm | −28.6 | −13.5 | −28.8 | −30.5 | −18.2 | −4.81 |
| | $h_j$ | cm | −9.23E4 | −9.43E4 | −7.48E4 | −7.87E4 | −1.97E4 | −2.63E4 |
| | $h_d$ | cm | −3.23E6 | −4.96E6 | −3.75E6 | −6.86E6 | −7.66E8 | −9.69E8 |
| VGN | $\theta_r$ | – | 5.25E–2 | 4.88E–2 | 4.48E–2 | 3.10E–2 | 1.60E–5 | 1.16E–6 |
| | $\theta_s$ | – | 0.472 | 0.491 | 0.461 | 0.540 | 0.501 | 0.379 |
| | $\alpha$ | cm$^{-1}$ | 1.62E–2 | 1.84E–2 | 1.53E–2 | 1.21E–2 | 2.62E–2 | 0.164 |
| | $n$ | – | 1.47 | 1.51 | 1.41 | 1.28 | 1.11 | 1.10 |
| VGA | $\theta_r$ | – | 5.27E–2 | 4.89E–2 | 6.12E–2 | 3.80E–4 | 7.21E–5 | 2.80E–5 |
| | $\theta_s$ | – | 0.472 | 0.491 | 0.457 | 0.540 | 0.500 | 0.379 |
| | $\alpha$ | cm$^{-1}$ | 1.62E–2 | 1.84E–2 | 1.49E–2 | 1.33E–2 | 3.66E–2 | 1.26 |
| | $n$ | – | 1.47 | 1.52 | 1.46 | 1.25 | 1.11 | 1.10 |
| | $h_{ae}$ | cm | −3.26E–3 | −2.08E–3 | −15.1 | −4.87 | −7.31 | −4.54 |
| RIA | $\theta_s$ | – | 0.474 | 0.493 | 0.463 | 0.540 | 0.500 | 0.379 |
| | $\alpha$ | cm$^{-1}$ | 2.04E–2 | 2.33E–2 | 1.86E–2 | 1.33E–2 | 3.57E–2 | 0.164 |
| | $n$ | – | 1.33 | 1.37 | 1.31 | 1.25 | 1.11 | 1.10 |
| | $h_{ae}$ | cm | −5.95E–3 | −2.36E–3 | −2.43E–3 | −4.80 | −7.12 | −1.21E–3 |
| | $h_j$ | cm | −1.49E6 | −1.07E6 | −8.62E6 | −8.02E6 | −8.33E6 | −9.89E6 |






Table C4. The fitting parameters and their values for six parameterizations for clayey soils from the
UNSODA database (National Agricultural Library, 2017; Nemes et al., 2001). The three–character
parameterization label is explained in the main text.

| Para–meteri–zation | Para–meter | Unit | Soil (UNSODA identifier and classification according to Twarakavi et al. (2010)) | | | | | |
|---|---|---|---|---|---|---|---|---|
| | | | 1135 C2 | 1182 C2 | 1122 C4 | 1123 C4 | 1180 C4 | 1181 C4 |
| BCO | $\theta_r$ | – | 3.94E–4 | 1.79E–4 | 2.64E–4 | 2.13E–4 | 5.22E–4 | 1.24E–5 |
| | $\theta_s$ | – | 0.420 | 0.549 | 0.362 | 0.358 | 0.497 | 0.456 |
| | $h_{ae}$ | cm | –106 | –0.977 | –10.0 | –10.0 | –11.1 | –5.17 |
| | $\lambda$ | – | 7.85E–2 | 4.41E–2 | 3.37E–2 | 2.69E–2 | 5.65E–2 | 5.40E–2 |
| FSB | $\theta_s$ | – | 0.420 | 0.548 | 0.360 | 0.356 | 0.495 | 0.456 |
| | $\theta_a$ | – | 0.400 | 0.307 | 0.350 | 0.340 | 0.491 | 0.345 |
| | $h_{ae}$ | cm | –106 | –0.230 | –5.75 | –10.0 | –8.58 | –13.2 |
| | $\lambda$ | – | 0.172 | 5.64E–2 | 6.60E–2 | 5.69E–2 | 100 | 8.08E–2 |
| RNA | $\theta_s$ | – | 0.420 | 0.549 | 0.370 | 0.370 | 0.497 | 0.456 |
| | $h_{ae}$ | cm | –106 | –3.61 | –9.99 | –9.99 | –0.150 | –7.66 |
| | $h_j$ | cm | –107 | –12.2 | –10.5 | –10.7 | –24.1 | –22.1 |
| | $h_d$ | cm | –1.64E8 | –1.00E9 | –1.00E9 | –1.00E9 | –1.00E9 | –1.00E9 |
| VGN | $\theta_r$ | – | 0.263 | 8.94E–6 | 1.16E–4 | 0.210 | 0.255 | 3.34E–6 |
| | $\theta_s$ | – | 0.413 | 0.548 | 0.359 | 0.354 | 0.496 | 0.456 |
| | $\alpha$ | cm$^{-1}$ | 1.02E–3 | 0.753 | 1.37E–2 | 2.92E–3 | 0.805 | 0.143 |
| | $n$ | – | 2.37 | 1.05 | 1.05 | 1.21 | 1.26 | 1.06 |
| VGA | $\theta_r$ | – | 0.263 | 1.19E–5 | 5.78E–2 | 0.188 | 2.63E–2 | 2.12E–6 |
| | $\theta_s$ | – | 0.413 | 0.548 | 0.359 | 0.354 | 0.498 | 0.456 |
| | $\alpha$ | cm$^{-1}$ | 1.02E–3 | 1.21 | 1.37E–2 | 3.22E–3 | 10.1 | 1.70 |
| | $n$ | – | 2.37 | 1.05 | 1.07 | 1.17 | 1.06 | 1.05 |
| | $h_{ae}$ | cm | –1.03 | –0.467 | –4.18E–2 | –10.0 | –1.11E–2 | –4.83 |
| RIA | $\theta_s$ | – | 0.416 | 0.548 | 0.359 | 0.354 | 0.497 | 0.457 |
| | $\alpha$ | cm$^{-1}$ | 1.86E–3 | 1.31 | 1.39E–2 | 4.00E–3 | 14.2 | 0.702 |
| | $n$ | – | 1.16 | 1.05 | 1.05 | 1.06 | 1.06 | 1.05 |
| | $h_{ae}$ | cm | –106 | –0.525 | –3.80E–3 | –9.99 | –4.41E–2 | –4.25 |
| | $h_j$ | cm | –7.57E6 | –9.97E6 | –9.56E4 | –4.43E6 | –415 | –206 |






Table C5. Root mean square errors of the parameter fits for the sandy or loamy soils (A1 and A2
soils according to Twarakavi et al., 2010)

| Parameterization | Soil (UNSODA identifier and classification according to Twarakavi et al. (2010)) | | |
| --- | --- | --- | --- |
| | 2126 A1 | 1142 A2 | 2104 A2 |
| BCO | 0.0620 | 0.0990 | 0.0481 |
| FSB | 0.0626 | 0.0990 | 0.0517 |
| RNA | 0.0659 | 0.0989 | 0.0553 |
| VGN | 0.0330 | 0.0252 | 0.0278 |
| VGA | 0.0330 | 0.0250 | 0.0278 |
| RIA | 0.0652 | 0.0504 | 0.0542 |





Table C6. Root mean square errors of the parameter fits for the sandy soils (A3 and A4 soils
according to Twarakavi et al., 2010)

| Parameterization | Soil (UNSODA identifier and classification according to Twarakavi et al. (2010)) | | | | | |
|---|---|---|---|---|---|---|
| | 1120 | 1143 | 2110 | 2132 | 1121 | 1133 |
| | A3 | A3 | A3 | A3 | A4 | A4 |
| BCO | 0.0926 | 0.0501 | 0.0445 | 0.0356 | 0.1288 | 0.0803 |
| FSB | 0.0926 | 0.0500 | 0.0445 | 0.0292 | 0.1054 | 0.0700 |
| RNA | 0.0926 | 0.0500 | 0.0457 | 0.0356 | 0.1286 | 0.0775 |
| VGN | 0.0446 | 0.0333 | 0.0378 | 0.0204 | 0.0720 | 0.0175 |
| VGA | 0.0489 | 0.0396 | 0.0445 | 0.0203 | 0.0720 | 0.0175 |
| RIA | 0.0643 | 0.0346 | 0.0491 | 0.0203 | 0.0720 | 0.0530 |







Table C7. Root mean square errors of the parameter fits for the silty soils.

| Parameterization | Soil (UNSODA identifier and classification according to Twarakavi et al. (2010)) | | | | | |
| | 3260 | 3261 | 3263 | 3250 | 3251 | 4450 |
| | B2 | B2 | B2 | B4 | B4 | B4 |
| --- | --- | --- | --- | --- | --- | --- |
| BCO | 0.0793 | 0.1316 | 0.0973 | 0.0822 | 0.0551 | 0.0499 |
| FSB | 0.0794 | 0.1316 | 0.0973 | 0.0815 | 0.0395 | 0.0445 |
| RNA | 0.0793 | 0.1316 | 0.0973 | 0.0822 | 0.0551 | 0.0499 |
| VGN | 0.0456 | 0.0607 | 0.0638 | 0.0413 | 0.0474 | 0.0485 |
| VGA | 0.0455 | 0.0607 | 0.0769 | 0.0412 | 0.0466 | 0.0497 |
| RIA | 0.0543 | 0.0698 | 0.0668 | 0.0412 | 0.0466 | 0.0485 |





Table C8.  Root mean square errors of the parameter fits for the clayey soils.

| Parameterization | Soil (UNSODA identifier and classification according to Twarakavi et al. (2010)) | | | | | |
| --- | --- | --- | --- | --- | --- | --- |
|  | 1135 | 1182 | 1122 | 1123 | 1180 | 1181 |
|  | C2 | C2 | C4 | C4 | C4 | C4 |
| BCO | 0.0913 | 0.0494 | 0.0349 | 0.0489 | 0.0187 | 0.0428 |
| FSB | 0.0721 | 0.0441 | 0.0212 | 0.0321 | 0.1196 | 0.0360 |
| RNA | 0.0812 | 0.0913 | 0.1235 | 0.1501 | 0.0347 | 0.0570 |
| VGN | 0.0208 | 0.0488 | 0.0197 | 0.0244 | 0.0411 | 0.0433 |
| VGA | 0.0208 | 0.0485 | 0.0198 | 0.0243 | 0.0197 | 0.0429 |
| RIA | 0.0519 | 0.0485 | 0.0197 | 0.0244 | 0.0180 | 0.0391 |







Table C9. Values for the hydraulic conductivity at saturation ($K_s$) for the selected soils from
the UNSODA database. The soils are identified by their UNSODA identifier. Their classification
according to Twarakavi et al. (2010) is also given.

| UNSODA identifier | Texture classification | $K_s$ (cm d$^{-1}$) |
| --- | --- | --- |
| 2126 | A1 | 1.10E3 |
| 1142 | A2 | 13.4 |
| 2104 | A2 | 553 |
| 1120 | A3 | 37.9 |
| 1143 | A3 | 23.5 |
| 2110 | A3 | 16.3 |
| 2132 | A3 | 5.52 |
| 1121 | A4 | 7.13 |
| 1133 | A4 | 7.13 |
| 3260 | B2 | 10.8 |
| 3261 | B2 | 32.0 |
| 3263 | B2 | 54.0 |
| 3250 | B4 | 1.51 |
| 3251 | B4 | 2.74 |
| 4450 | B4 | 1.20 |
| 1135 | C2 | 0.142 |
| 1182 | C2 | 1.25 |
| 1122 | C4 | 2.92 |
| 1123 | C4 | 0.740 |
| 1180 | C4 | 215 |
| 1181 | C4 | 178 |






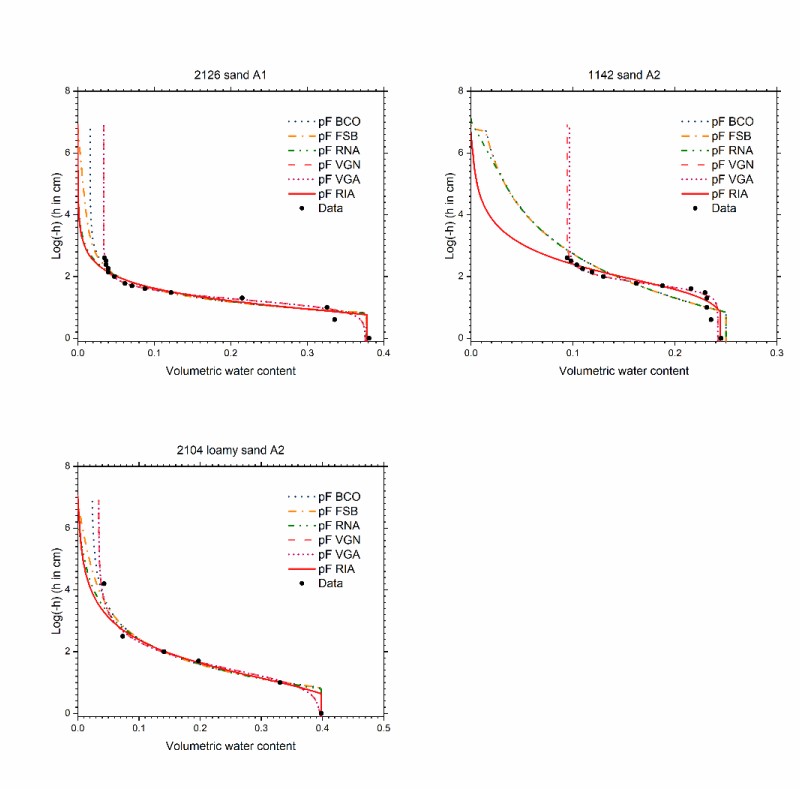


Figure C2. Retention data and fitted soil water retention curves according to six

parameterizations for selected UNSODA soils with Twarakavi et al.'s (2010) A1 or A2

classification. The parameterizations are explained in the text.



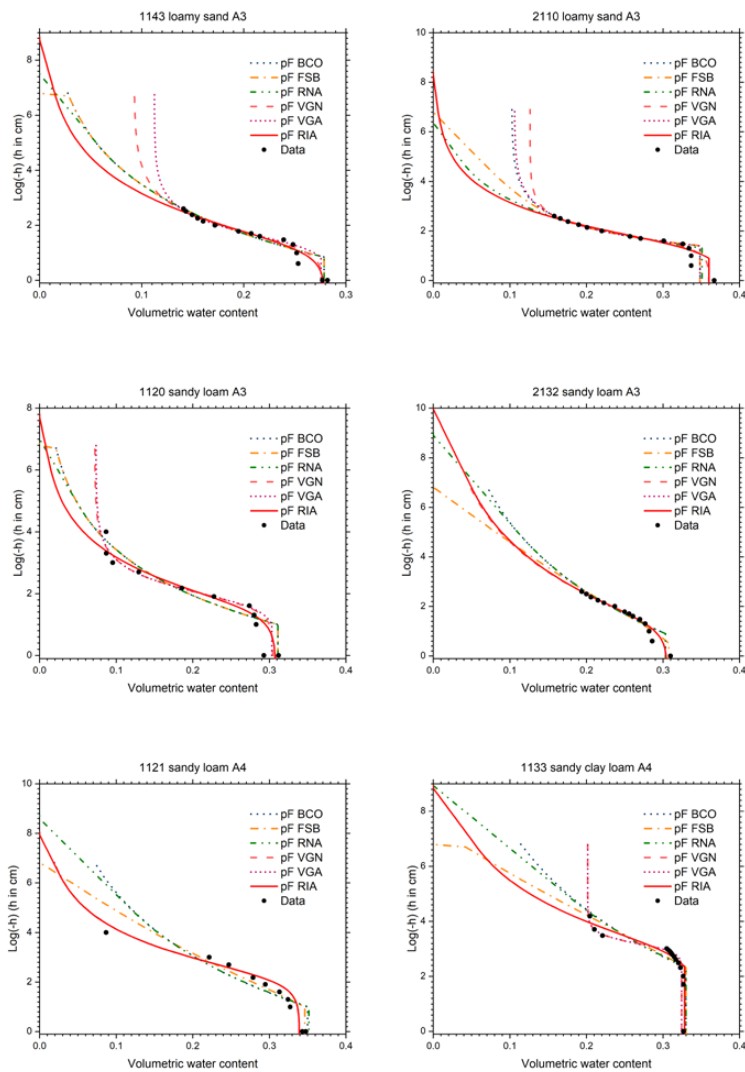


Figure C3. Retention data and fitted soil water retention curves according to six

parameterizations for selected UNSODA soils with Twarakavi et al.'s (2010) A3 or A4

classification. The parameterizations are explained in the text.



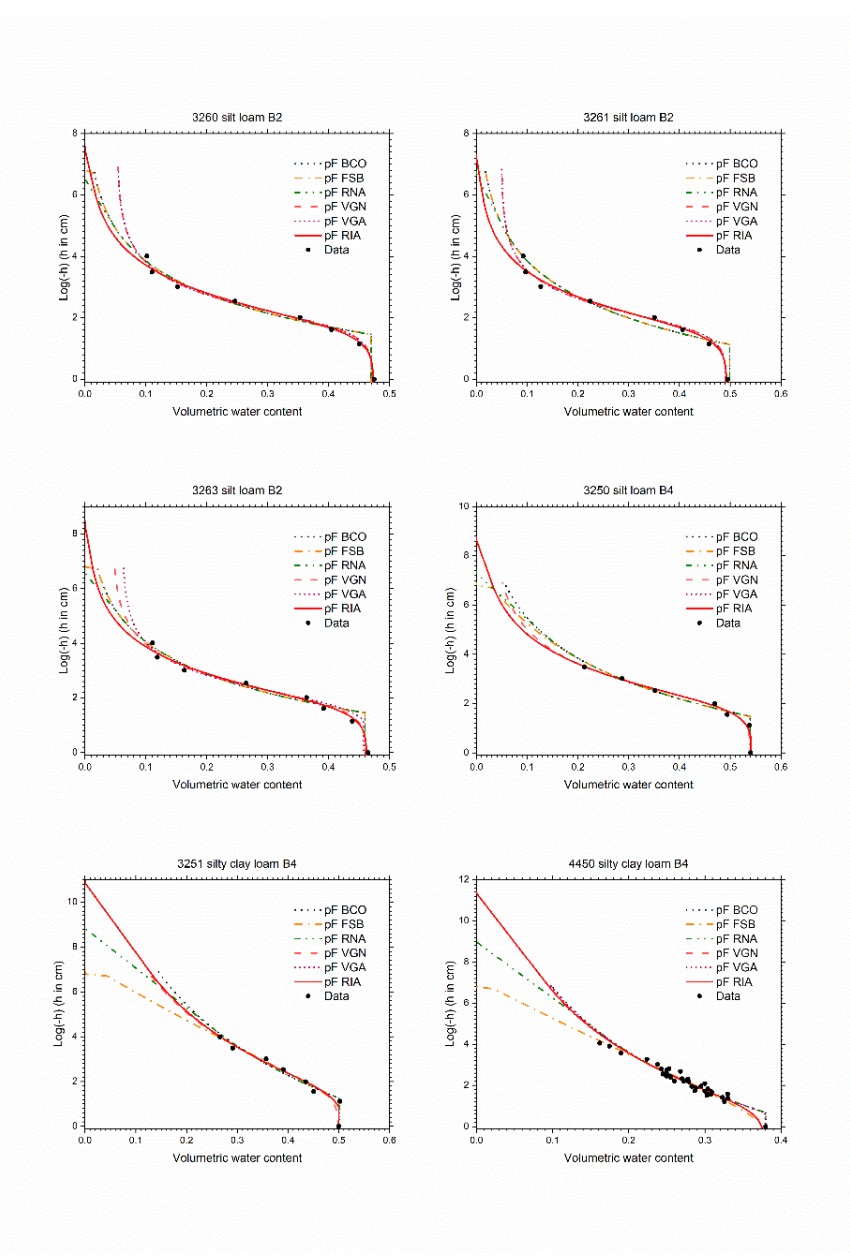


Figure C4. Retention data and fitted soil water retention curves according to six
parameterizations for selected UNSODA soils with Twarakavi et al.'s (2010) B2 or B4
classification. The parameterizations are explained in the text.



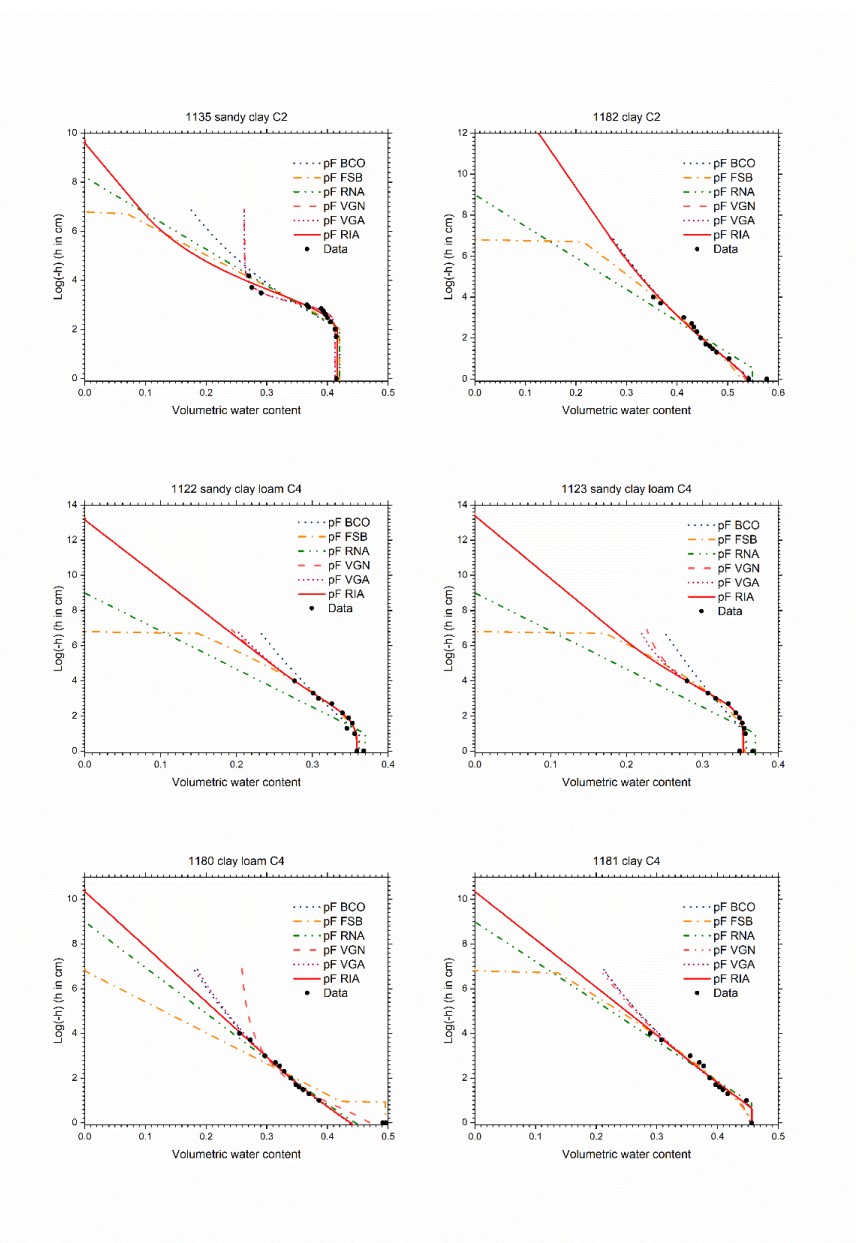


Figure C5. Retention data and fitted soil water retention curves according to six

parameterizations for selected UNSODA soils with Twarakavi et al.'s (2010) C2 or C4

classification. The parameterizations are explained in the text.



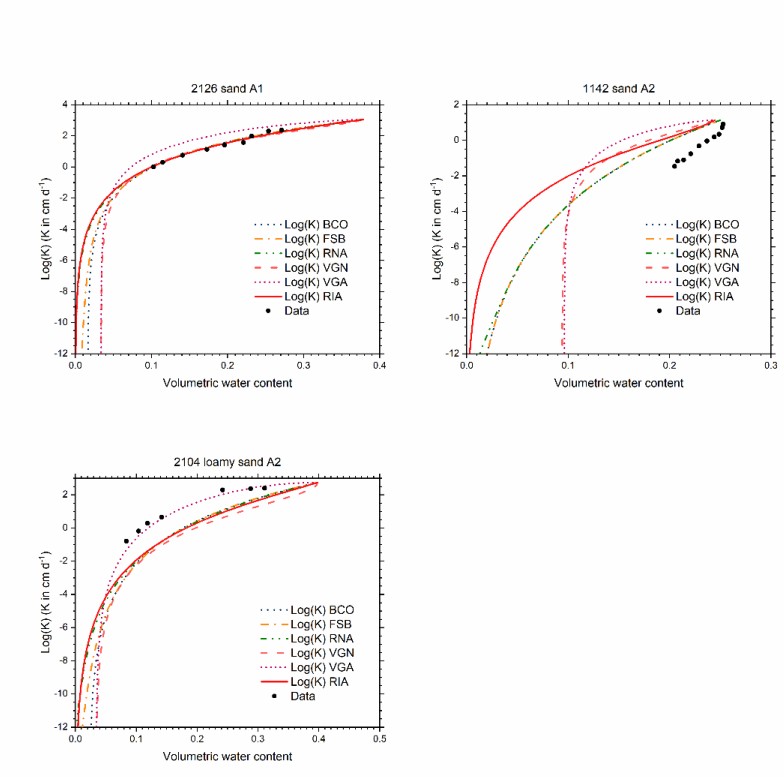


Figure C6. Conductivity data and conductivity curves derived from the retention curves of
Fig. C2 according to six parameterizations for selected UNSODA soils with Twarakavi et al.'s
(2010) A1 or A2 classification.



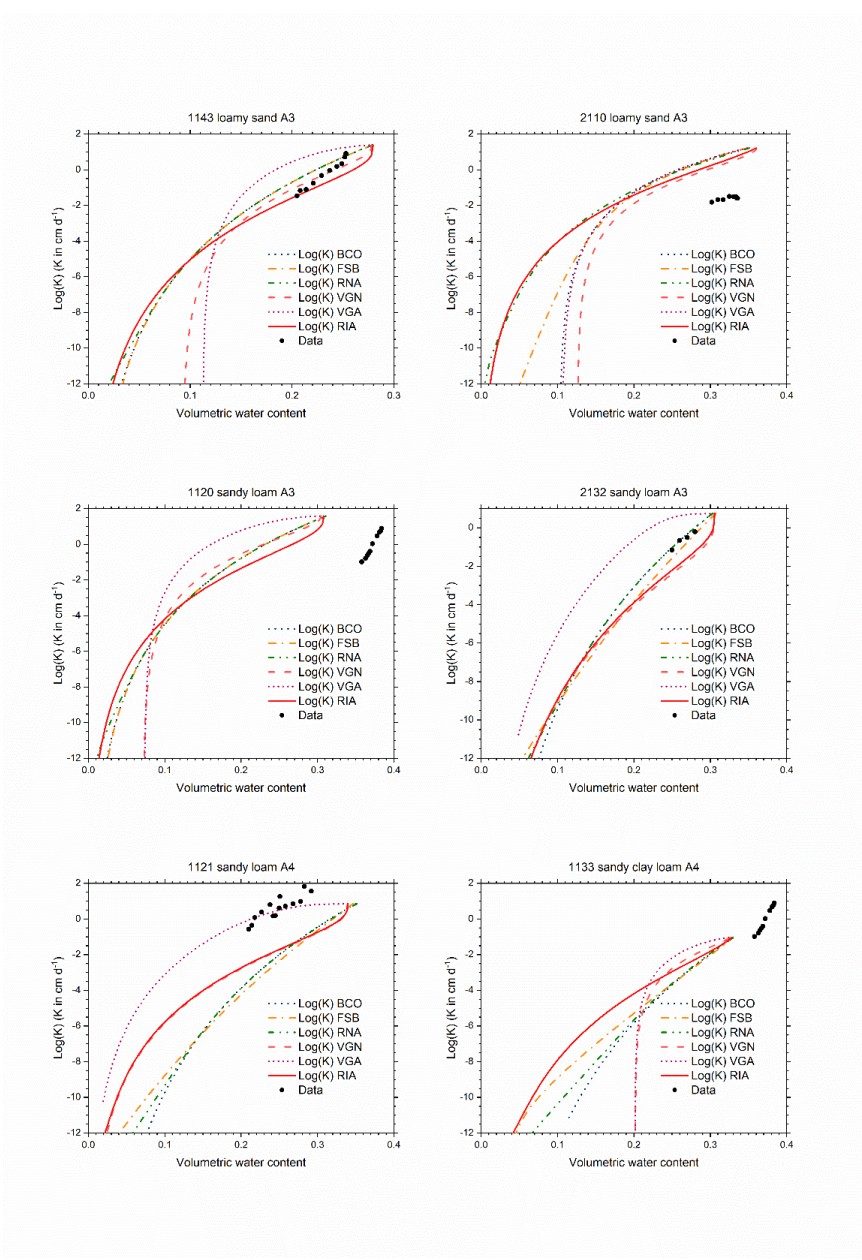


Figure C7. Conductivity data and conductivity curves derived from the retention curves of

Fig. C3 according to six parameterizations for selected UNSODA soils with Twarakavi et al.'s

(2010) A3 or A4 classification.



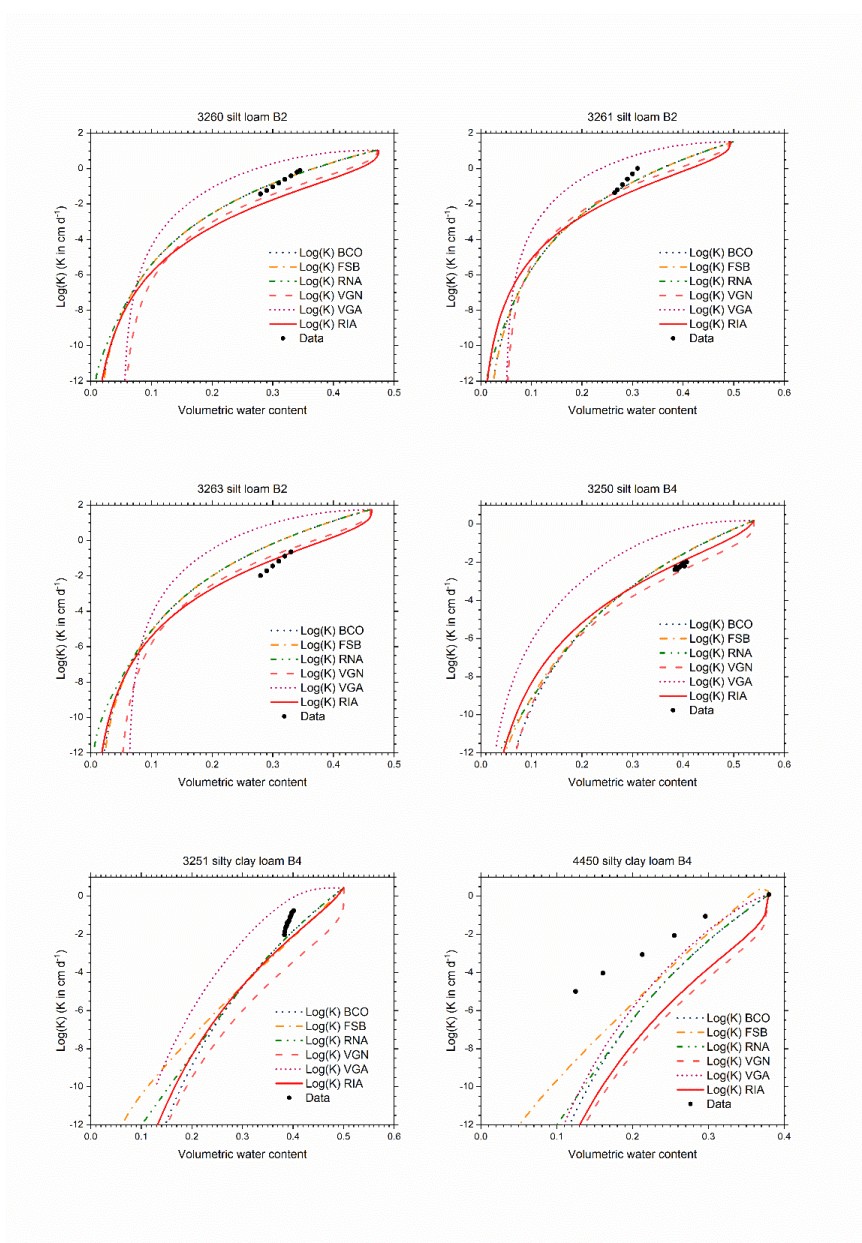


Figure C8. Conductivity data and conductivity curves derived from the retention curves of
Fig. C4 according to six parameterizations for selected UNSODA soils with Twarakavi et al.'s
(2010) B2 or B4 classification.



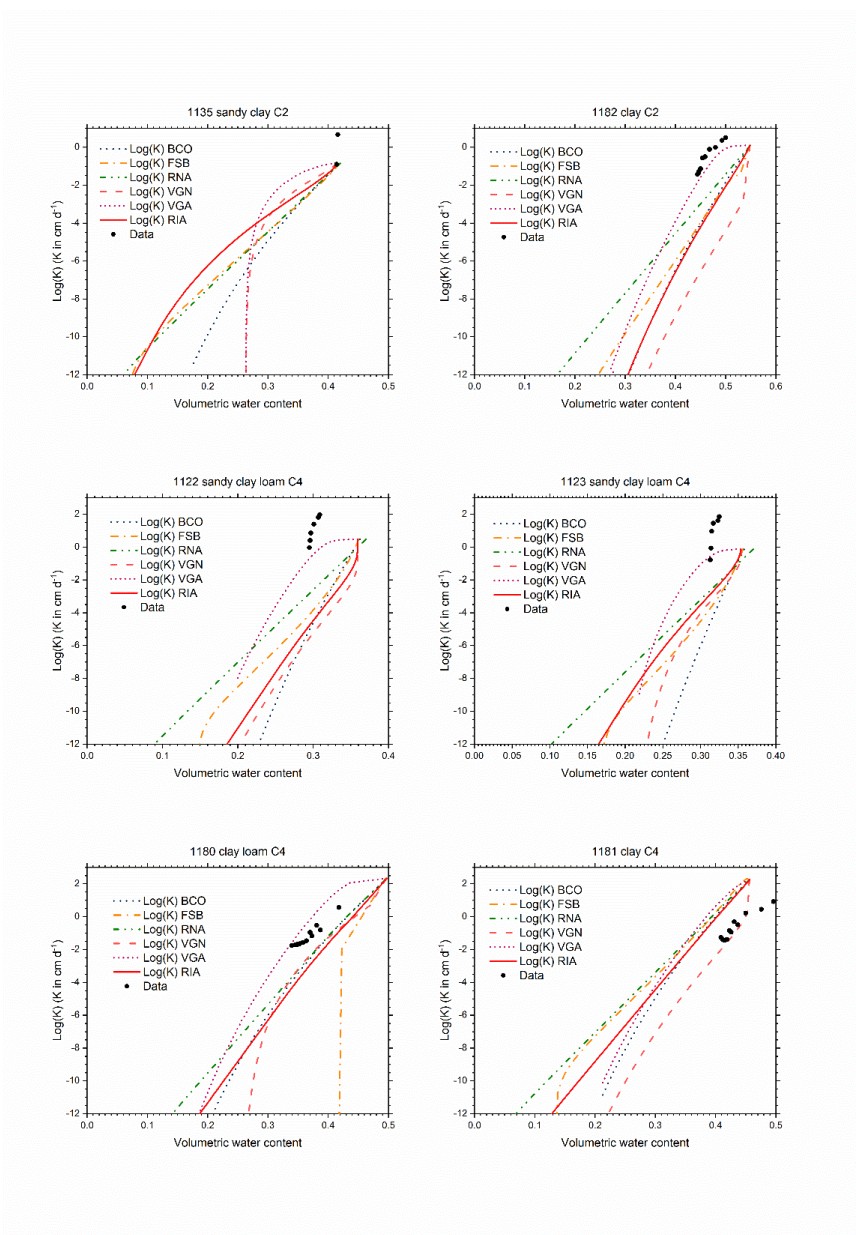


Figure C9. Conductivity data and conductivity curves derived from the retention curves of

Fig. C5 according to six parameterizations for selected UNSODA soils with Twarakavi et al.'s

(2010) C2 or AC4 classification.




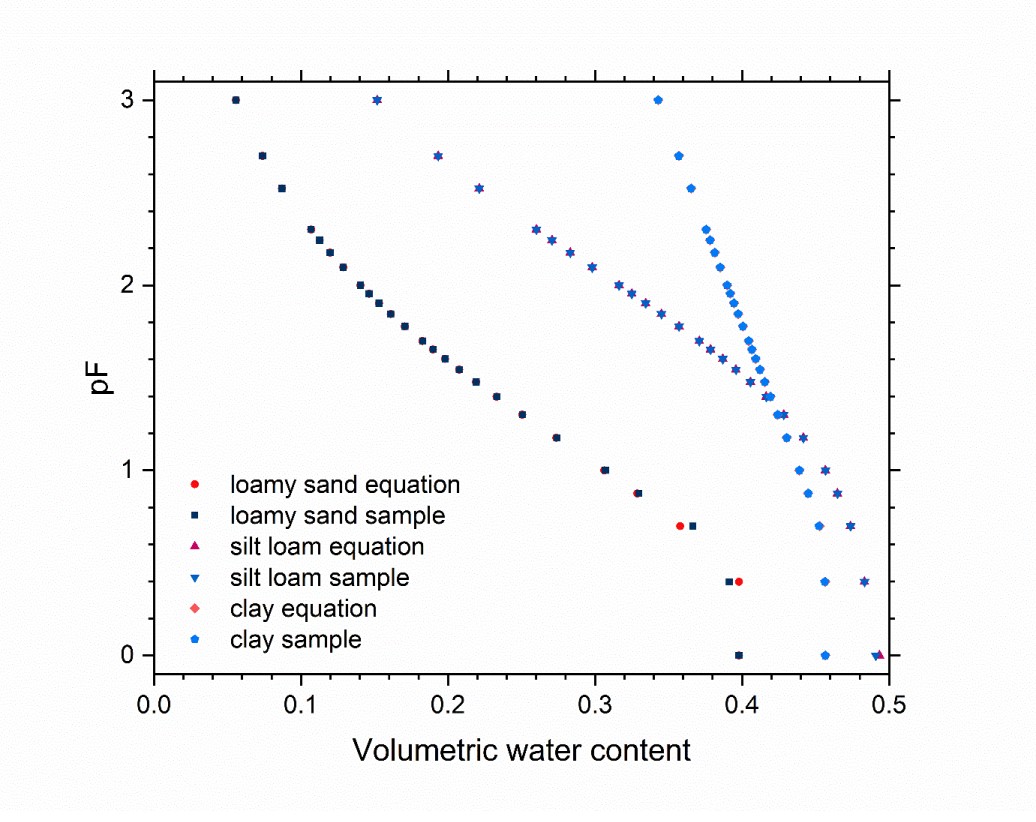


Figure C10. Soil water retention points calculated from RIA parameterizations for loamy

sand (UNSODA identifier 2104, classification according to Twarakavi et al. (2010) A4), silt

loam (3261, B2), and clay (1181, C4). The points were either calculated directly for the

given pF value ('equation'), or by calculating the average water content in a sample of 5.0

cm height at hydrostatic equilibrium, with the matric potential at the center of the sample

corresponding to the indicated pF value ('sample'). N.B. The data points for zero matric

potential were plotted at pF = 0 instead of pF = −∞.




## Author contributions



GdR developed the new parameterization, assisted by RM. GdR identified the 21 soils from
the UNSODA database and designed the model test. JM developed the SCE code for
parameter identification. GdR coded the shell around the SCE core to make it suitable for
fitting various SWRC parameterizations. GdR collected the weather data and generated the
weather records. GdR and RM carried out the paramter fitting and the Hydrus–1D
simulations. GdR wrote the manuscript with contributions from JM and RM.

## Competing interests


GdR is a member of the HESS Editorial Board.



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
