# Peer review of "Sigmoidal Water Retention Function with Improved Behavior in Dry"

_Hydrology and Earth System Sciences, 2020_

## Referee Comment (RC1) · Anonymous Referee #1 · 4 Oct 2020

In this manuscript, the authors report some new formulations for the soil water retention curve and hydraulic conductivity curve. The formulations work at different moisture conditions, from the dry condition to the fully saturated condition. The manuscript is generally well written and can be followed easily. Prior to a final recommendation, the authors should address the following comments. 1. In the literature, some existing formulations have comparable capabilities with the newly proposed formulations. The authors should explain the new contribution clearly. 2. Equation (4) plays a very important role in the manuscript, but there is no sufficient discussion about this equation. For example, how is this equation derived? What are its advantages compared to existing equations? 3. For many figures, the curves cannot be differentiated easily if printed in black and white.

---

## Referee Comment (RC2) · Anonymous Referee #2 · 9 Oct 2020

GENERAL COMMENTS The standard parameterization of soil hydraulic functions that is used in the modeling of unsaturated water flow may imply 'non-physical' behavior for certain parameter combinations (i.e. soil types). The authors propose a new approach with better description of the processes under very dry and very wet conditions. Based on numerical experiments, the authors could show that the simulations using the new proposed hydraulic functions were more stable compared to other parameterizations (simulations could be completed for more scenarios with fine textured soils). The discussion of the limits of the standard approach is important to ensure that it is not applied in an uncritical way. In addition, the parameterization of hydraulic functions allowing stable simulations for a wide range of soil types and conditions is relevant. Hence, the motivation and objective of the paper are good but I'm questioning (i) the

model interpretation and the testing based on (ii) comparison with measured hydraulic properties and (iii) numerical experiments using Hydrus 1D. In short, I propose to use different data sets for model comparison and a more detailed discussion of representing flow processes under dry conditions.

A) In general, none of the discussed models matches the measured unsaturated hydraulic conductivity data (as confirmed by the authors in lines 318-319). The authors hypothesize that these differences between model and measurements are a result of the contrast between measuring soil water retention (and saturated conductivity) in the lab and unsaturated hydraulic conductivity in the field. I agree that conducting measurements in the field will have a big effect and introduce uncertainties - but because the comparison between hydraulic conductivity functions is in the core of this paper, the comparison must be done with measurements that do not have this lab/field-problem. The authors should look for a few measurements from different soil types with very reliable and consistent measurements of both unsaturated conductivity and water retention done in the lab. Specifically, instead of choosing data from UNSODA data base, it would be important to select measurements from papers that are measuring both properties in consistent systems (lab studies).

B) Related to the problem of inconclusive comparison with experimental unsaturated hydraulic conductivity data is the comparison with numerical simulations between the "VGA" model by Ippisch et al. and the new RIA-model. The results are very sensitive to differences in hydraulic functions at high and intermediate water contents and depend on the accuracy of matching the unsaturated hydraulic conductivity in this water content range. The authors should explain in more detail the differences between the conductivity function of VGA and RIA and – based on comparison with high quality measurements – which approach may be more appropriate.

C) The authors state that the 'behavior' at the dry end with water content dropping to zero for finite pressure values is more appropriate. I agree that at some point even the last molecular layer of adsorbed water will be removed - but can the flow processes

under such conditions be described properly with the hydraulic functions proposed in this paper? Are the appropriate physical processes under dry conditions (film flow, vapor transport, ....) described properly with eq. (12) and (13)? As far as I understand, the hydraulic conductivity functions used in RIA (and VGA and VGN) are based on eq. (12) but this expression relies on capillary flow and does not reproduce the physics of film flow (or vapor transport). So, the simulations at the very dry end – that should reproduce the dynamics of water adsorption and film flow – are based on equations valid for capillary flow. The authors should comment on that.

D) Similar to the discussion of the hydraulic properties at the wet end, the authors should compare the model with measurements (also of K(theta)) at very negative pressure levels.

SPECIFIC QUESTIONS I'm questioning the choice of the title using the terms 'Improved behavior in dry and wet soils' because it was not shown that the 'behavior' was improved.

Lines 58-62: Please expand this paragraph by 1-2 sentences to explain how the water uptake capacity becomes unlimited.

Lines 105-117: The authors should expand on the physical differences between adsorption and capillary forces. The entire paragraph is on water retention only and not on water flow under such dry conditions. The authors must discuss different types of flow related to film and corner flow and how this could be implemented (see Tuller and Or, Hydraulic conductivity of variably saturated porous media: Film and corner flow in angular pore space, Water Resources Research, 37, 1257-1276, 2001)

Line 161, eq. (5): I understand that the value of parameter beta is determined using eq. (7). I would have expected that beta should depend on the surface area of the soil (determining the amount of adsorbed water). Could the authors please comment on that?

Line 209, eq. (13): Is tau chosen as 0.5?

Lines 225 – 231: I propose to choose less but better data with (i) consistent lab measurements for both SWRC and K(theta) and (ii) some data with K(theta) values at very negative pressures

Figure 1: Why RIA and VGA are different for the loamy sand?

Table 2: You should add Ks and hd (and beta) values for RIA

Figure 2: Choose different color for VGA for silt loam – it is too similar to clay

Lines 293 - 298: The reference to Bitelli and Flury is illustrative – maybe the authors could use some of those data as well to fit SWRC

Line 320: the only sample with K(theta) measured in the lab (soil 4450) is poorly described by RIA . . .

Lines 331-335: There is no experimental evidence that the RIA trends are better – this is just a description of modeled behavior

Figure C5 & C9, Soil 1122 (and other examples): The authors obviously cut the curves at pF=6.8 for VGA and VGN. Probably this should be stated and explained in the captions

Lines 363-365: The statement that 'small differences between SWRCs can have a significant influence through different hydraulic conductivity curves' should probably be revised; even for the very same SWRC curve the water flow will be different due to different conductivity functions

Lines 385-386: The statement that 'RIA was better able to produce a conductivity curve with a substantial drop . . .' is misleading because we do not know if this 'substantial drop' is in agreement with measurements

TECHNICAL CORRECTIONS/COMMENTS Line 17: State that the infinite slope at

saturation is considered to be physically impossible

Line 37: Write out SWRC at the beginning of the main text

Line 96: the shape of the samples ('cylindrical') is not relevant; maybe 'equilibrating short soil samples'?

Line 157: delete 'the' ('... the the logarithmic ...)

Line 238: Is Tamale not in the tropical climate zone?

Line 315: delete 'goes'

Line 344-345: what do you mean with 'reduced Ks-value' and the 'high Ks-value' for clay soils?

Line 352: for the loamy sand under temperate climate, the results ranged from 92 to 104 % (Table 3) – why is this difference more than 10%? Was the deviation in the silt loam (84-115%) not higher?

Line 353: For loamy sand in 'semi-arid' climate, the 89% value for evaporation has higher deviation than 10%

Line 354: Temperate, not 'temperature'

Line 439: delete 'of'

---

## Author Comment (AC1) · 13 Oct 2020

HESS 2020-380. Sigmoidal Water Retention Function with Improved Behavior in Dry and Wet Soils.

Response to referee 1

The original review report:

In this manuscript, the authors report some new formulations for the soil water retention curve and hydraulic conductivity curve. The formulations work at different moisture conditions, from the dry condition to the fully saturated condition. The manuscript is generally well written and can be followed easily. Prior to a final recommendation, the authors should address the following comments. 1. In the literature, some existing

formulations have comparable capabilities with the newly proposed formulations. The authors should explain the new contribution clearly. 2. Equation (4) plays a very important role in the manuscript, but there is no sufficient discussion about this equation. For example, how is this equation derived? What are its advantages compared to existing equations? 3. For many figures, the curves cannot be differentiated easily if printed in black and white.

Below, we address the points raised by the referee.
* * *
Referee:

1. In the literature, some existing formulations have comparable capabilities with the newly proposed formulations. The authors should explain the new contribution clearly.

2. Equation (4) plays a very important role in the manuscript, but there is no suffcient discussion about this equation. For example, how is this equation derived? What are its advantages compared to existing equations?

Response:

The review is brief but constructive. Upon careful reading, we believe points 1 and 2 raised by the referee are related and can best be treated simultaneously. If we are permitted to revise the paper we intend to address these two points as follows:

In the Introduction in its current form we outline two lines of research: one devoted to the problems of the van Genuchten (1980) curve at the wet end, and one devoted to the problems that arise when a retention curve has an asymptote at the dry end (as the van Genuchten curve has). The introduction gives an equation (Eq. [3] that solves the problem at the wet end, but does not provide an equation that eliminates the problem in the dry end (the junction model of Rossi and Nimmo, 1994).

We can modify the Introduction by providing the equation of Rossi and Nimmo also.

The Introduction then proceeds as it does in its current form, by stating that we are going to combine both equations.

The Theory section shows the combination of both equations announced in the Introduction. When looking at Eq. [3] and Rossi and Nimmo's equation, it is easy to see how Eq. [4] arises from them. We will add some text that explains which elements of both equations were adopted and how we combined them, so that the origins and derivation of Eq. [4] are clarified (point 2 of the referee) and we can explain that the earlier equations only solved one of two problems (point 1 of the referee).

The advantages of the new equation are:

- the reciprocal of the slope is d theta / d h. At saturation, its value is zero, which stabilizes the hydraulic conductivity near saturation.

- no asymptote at the dry end, which keeps the area under the retention curve finite

- a sigmoidal shape, which allows an extension to a multimodal form

- more robust than previous van Genuchten-based sigmoidal curves when used in numerical simulations
* * *
Referee:

3. For many figures, the curves cannot be differentiated easily if printed in black and white.

Response:

This problem only pertains to the figures in Appendix C. The graphs in that appendix serve to give the reader an idea about the shape of the curves corresponding to the parameter values provided, and we therefore expect that consultation of the figures on the screen (in color) will often suffice. In several cases there are only small differences

in the shape of the curves of different parameterizations, so they (nearly) overlap. In these cases, readers still have a good idea of the shape of their curve of choice. The tables with parameter values and the graphs of the corresponding curves in the appendix are provided as a service for readers who need soil hydraulic properties for areas where there is limited information available, or who need representations for broad soil categories. For these purposes, overlapping curves do not pose a problem. The idea is that readers choose a soil type and a parameterization and then use the corresponding parameters in their calculations. They will probably create their own graph for their particular parameter set.
* * *
On behalf of all authors,

Gerrit de Rooij
* * *

---

## Author Comment (AC2) · 15 Oct 2020

Sigmoidal Water Retention Function with Improved Behavior in Dry and Wet Soils HESS 2020-380

Reply to referee 2

The text by the referee is in *blue and italics*. Our response is in black and regular font.

Referee report

*GENERAL COMMENTS The standard parameterization of soil hydraulic functions that is used in the modeling of unsaturated water flow may imply 'non-physical' behavior for certain parameter combinations (i.e. soil types). The authors propose a new approach with better description of the processes under very dry and very wet conditions. Based on numerical experiments, the authors could show that the simulations using the new proposed hydraulic functions were more stable compared to other parameterizations (simulations could be completed for more scenarios with fine textured soils). The discussion of the limits of the standard approach is important to ensure that it is not applied in an uncritical way. In addition, the parameterization of hydraulic functions allowing stable simulations for a wide range of soil types and conditions is relevant. Hence, the motivation and objective of the paper are good but I'm questioning*

*(i)    the model interpretation and the testing based on*
*(ii)   comparison with measured hydraulic properties and*
*(iii)  numerical experiments using Hydrus 1D.*

*In short, I propose to use different data sets for model comparison and a more detailed discussion of representing flow processes under dry conditions.*

*A) In general, none of the discussed models matches the measured unsaturated hydraulic conductivity data (as confirmed by the authors in lines 318-319). The authors hypothesize that these differences between model and measurements are a result of the contrast between measuring soil water retention (and saturated conductivity) in the lab and unsaturated hydraulic conductivity in the field. I agree that conducting measurements in the field will have a big effect and introduce uncertainties - but because the comparison between hydraulic conductivity functions is in the core of this paper, the comparison must be done with measurements that do not have this lab/field-problem. The authors should look for a few measurements from different soil types with very reliable and consistent measurements of both unsaturated conductivity and water retention done in the lab. Specifically, instead of choosing data from UNSODA data base, it would be important to select measurements from papers that are measuring both properties in consistent systems (lab studies).*

Reply:
The referee states that the comparison between hydraulic conductivity functions is in the core of this paper, but we intended to focus on the retention curve, as we make clear in the title and the abstract. We recognize that the conductivity curve is important as well, but only devote attention to the effect of the retention curve parameter on the conductivity curve. For a more in-depth treatment of the conductivity curve we refer to Weber et al.

The referee argues we should seek lab data of soil water retention and conductivity measured on a consistent system. The underlying thought appears to be that laboratory data are inherently superior to field data. For applications to practical problems (that by necessity arise in the field) this is not necessarily the case, particularly for conductivity data. Several colleagues with considerable experience in field work have grown rather critical of applying lab-measured conductivities in the field because they often do not perform well. One reason may be that samples that are adequate for the volumetric water content are too small for the hydraulic conductivity: the representative elementary volume for the unsaturated hydraulic conductivity may be larger than the 100 cm$^3$ often used for soil water retention measurements. Larger samples (1000 cm$^3$ or more) appear to give more consistent results from anecdotal evidence, but then we do not have the consistent system that the referee prefers.

Based on our fits as well as on earlier reports of conductivity curves based on parameters fitted on soil water retention data, we think that it is better to fit a separate set of conductivity parameters on conductivity data when possible. In this paper, however, we wanted to compare various retention curves. Had we used all three parameterizations with the same conductivity curve, there would have been the risk that the conductivity curve would have had a dominating effect the simulation results. Also, we would not have been able to demonstrate the surprisingly large effect of the choice of the parameterization on the conductivity curve. For this particular goal, our choice was suitable.

We compared the estimated conductivity curves with independent data to present the complete picture to the readership, in order to encourage a critical and sensible use of our new parameterization. If a reader concludes that RIA is a useful full-range retention curve but prefers to independently fit a conductivity curve, we conveyed our message succesfully. This appears to be the case with referee 2. We can try to modify the text to bring this message across more clearly.

*B) Related to the problem of inconclusive comparison with experimental unsaturated hydraulic conductivity data is the comparison with numerical simulations between the "VGA" model by Ippisch et al. and the new RIA-model. The results are very sensitive to differences in hydraulic functions at high and intermediate water contents and depend on the accuracy of matching the unsaturated hydraulic conductivity in this water content range. The authors should explain in more detail the differences between the conductivity function of VGA and RIA and – based on comparison with high quality measurements – which approach may be more appropriate.*

Reply:
We do not consider the comparison with independently measured conductivities inconclusive. Our message that retention curve parameters do not describe the hydraulic conductivity curve very well came across, but apparently was not perceived as such. The limited transferability of retention parameters to conductivity curves has been known for several decades, so we are not breaking new ground here. We need to try to improve the clarity of our text to communicate this message more clearly.

Independently of this, the shape of the conductivity curve that arises from the RIA parameterization gave quite different and more plausible simulation results than VGA, and we reported this. This comment is related to this particular finding.

Generally speaking, one should fit a separate set of parameters directly to conductivity data when these are available (see above). If one has to rely on retention parameters only, RIA gives a numerically robust set of soil hydraulic property curves, but the simulation results should be applied with care if they are used in practical applications. The referee suggests that we test the ability of the retention parameters to describe the conductivity curve. We hesitate to do so because it has already been established that this does not work very well and we are not challenging this. The referee essentially hypothesizes that retention curve parameters can describe the conductivity curve, but we believe this hypothesis to be false. Such a test would also drift away from the main contribution of the paper, which is to introduce a sigmoidal retention curve without unphysical behavior in the dry and the wet end.

We read our text with the interpretation of the referee in mind. We realized that we unconsciously assumed that the readers shared our views on the limited usefulness of retention parameters for conductivity curves. But for a more optimistic reader like referee 2, our comparison of estimated and measured conductivities is indeed disappointing, and our appreciation of the ability of the estimated curves to sometimes correctly reproduce the shape of the conductivity curve can be a bit bewildering. We thank the referee for bringing this alternative viewpoint to our attention so vividly. We will take into consideration a broader spectrum of opinions on this matter when we rewrite the text in order to avoid unintendingly raising expectations.

The referee mentions the need for high-quality data to perform the test. Much of the data in the UNSODA data base is quite good, and in sheer size and range of soil types, the data base is unparalleled. More importantly, the referee may be too optimistic about the data quality that can be achieved. Unsaturated conductivity data tend to be quite noisy and not always transfer to field conditions well. We distinguish three causes for this:

- the need to measure matric potentials at two heights in the sample that are only centimeters apart. Especially in conductive soils, this leads to substantial errors in the estimates of the matric potential gradient.
- the difficulty with accurately measuring very low fluxes in the dry range. Even when one avoids the need to measure matric potentials below the tensiometer range by imposing matric potentials at opposite sides of the sample, one still has to wait days or more before sufficient leachate is collected to have a valid measurement. If steady state conditions are required this adds days or weeks to the procedure. These long times increase the risk of growth of bacteria or fungi in the sample and the porous plates that are probably needed on the inlet and outlet side of the sample. Furthermore, the small amounts of leachate in combination with the long time intervals make the data collection vulnerable to water losses from diffusion through tube walls or evaporation of the collected leachate from the collection vessel.
- the sample may be too small for measuring hydraulic conductivity. The representative elementary volume may be larger than the sample, and the flow lines in a sample may be forced to be more unidirectional than they are *in situ*.

This poses severe hurdles to carrying out the analysis the referee proposes. Newer setups (such as the HyProp apparatus) produce an unprecedented number of data points (but not in the dry range) at the cost of a relatively small sample size, leading to the size-related problems of the final bullet point.

Given our reservations about using the retention curve parameters to estimate the conductivity curve, we opted for a comparative analysis in the paper and report the main differences between RIA and VGA: the generally faster drop for RIA vs. VGA in the conductivity when the soil becomes slightly unsaturated, and the resulting gradual response of the flux at 2 m depth to rainfall for RIA vs. the rapid and jumpy response of VGA. We are not sure there is an added benefit of a very detailed analysis of a conductivity curve that the referee criticizes below for being insufficiently sophisticated anyway.

We also point out that we present fits to data sets of over 20 soils. In comparison to many of the other papers that introduce parametric expressions of soil water retention curves this is a large number. Only one of these papers included a comparative evaluation of their parameterization by using it in a numerical model, which is the main application of such parameterizations. We therefore already did more work than most to test the performance of our parameterization.

*C) The authors state that the 'behavior' at the dry end with water content dropping to zero for finite pressure values is more appropriate. I agree that at some point even the last molecular layer of adsorbed water will be removed - but can the flow processes under such conditions be described properly with the hydraulic functions proposed in this paper? Are the appropriate physical processes under dry conditions (film flow, vapor transport, ....) described properly with eq. (12) and (13)? As far as I understand, the hydraulic conductivity functions used in RIA (and VGA and VGN) are based on eq. (12) but this expression relies on capillary flow and does not reproduce the physics of film flow (or vapor transport). So, the simulations at the very dry end – that should reproduce the dynamics of water adsorption and film flow – are based on equations valid for capillary flow. The authors should comment on that.*

Reply:
The observation by the referee that the conductivity curve we used is based on capillary flow is correct. We are currently pursuing a way to link improved conductivity curves (based on the papers the referee quotes and other work) to RIA, but we ran into a theoretical issue with the current crop of available curves that has not received attention to our knowledge. We are still working on that. Given the amount of new material (theoretical and mathematical) that we are developing in that project, we concluded that adding it to the work reported here would result in a long, confusing paper with too many lines of thought.

In very dry conditions, vapor flow becomes significant and it can even exceed liquid water flow. When isolated pockets with liquid water and soil air are both present, water moves at least in part by evaporating from one pocket of liquid water and condensing in another, leading to trains of evaporation-condensation sequences. This can only be roughly modeled by a continuum model like Richards' equation through the use of

effective parameters and is not represented by any model of strictly liquid flow, be it capillary flow, film flow, corner flow, or a combination of those. The currently available models for film flow and corner flow do not fully capture the architecture of the pore space. They are an improvement over strictly capillary models, but not yet the definite representation of water dynamics in dry porous media.  Equations (12) and (13) are definitely less valid for dry conditions than for wet conditions, but more elaborate conductivity models only offer a partial improvement.

The referee requested us to comment on this, so adding a brief discussion below Eq. (13) of the issues outlined above would adequately address this comment. The effect of these processes on the relative magnitude of the fluxes in dry soils may well be very large, but because the fluxes in dry periods are orders of magnitude smaller than those in wet periods, their effect on the soil water balance over longer periods will generally be quite small.

*D)  Similar to the discussion of the hydraulic properties at the wet end, the authors should compare the model with measurements (also of K(theta)) at very negative pressure levels.*

Reply:
This is easier said than done, as can be seen from our discussion above of the limitations on measuring hydraulic conductivities in dry soils. We can measure neither matric potentials below the tensiometer range in soil samples, nor low fluxes at sufficient accuracy. In recent years, sensors that measure matric potentials down to wilting point have become available commercially (https://www.ugt-online.de/en/products/soil-science/tensiometers/full-range-tensiometer/), but they cannot (yet?) be miniaturized to a scale that would make them useful for hydraulic conductivity measurements. We doubt that the data the referee would like us to use will become available in the near future.

There are virtually no water content measurements reported for pF values above 4.2. Measurements in the dry range are limited, and, as Bitelli and Flury showed, often overestimate the water content. We can signal this problem in the paper, but not solve it.

*SPECIFIC QUESTIONS*
*I'm questioning the choice of the title using the terms 'Improved behavior in dry and wet soils' because it was not shown that the 'behavior' was improved.*

Reply:
Our parameterization is the only sigmoidal parameterization with a finite slope at saturation and a finite matric potential at zero water content. Both of these are improved behaviors. The mathematical evidence is provided. The referee may have interpreted 'behavior' as 'performance', but the performance, insofar we were able to determine it, improved as well.

*Lines 58-62: Please expand this paragraph by 1-2 sentences to explain how the water uptake capacity becomes unlimited.*

Reply:
The current paragraph reads:

Fuentes et al. (1991) warned that the asymptotic residual water content at the dry end could lead to a non–converging integral of the retention curve, and showed how this would mathematically lead to a physically impossible unlimited water uptake capacity of a finite soil column. From their analysis follows that this can only be prevented if $n > 2$ in Eq. (1), a condition which is often not satisfied.

We propose to replace it by:

Fuentes et al. (1991) warned that the asymptotic residual water content at the dry end could lead to a non–converging integral of the retention curve when the integration is carried out between the saturated water content and a water content that approximates the residual water content in the limit. In that case, the area below the retention curve becomes infinite. Fuentes et al. (1991) showed that this would lead to an unlimited amount of water stored in a column of a finite length at hydrostatic equilibrium if its height was such that the residual water content was approximated closely at the top of the column. This physically impossible case is only avoided if $n > 2$ in Eq. (1), a condition which is often not satisfied.

*Lines 105-117: The authors should expand on the physical differences between adsorption and capillary forces. The entire paragraph is on water retention only and not on water flow under such dry conditions. The authors must discuss different types of flow related to film and corner flow and how this could be implemented (see Tuller and Or, Hydraulic conductivity of variably saturated porous media: Film and corner flow in angular pore space, Water Resources Research, 37, 1257-1276, 2001)*

Reply:
The paper focuses on the retention curve, and therefore the paragraph in question dealt with water retention. The conductivity curve is only addressed in terms of the way it is affected by the retention curve parameters in order to explain the significant differences of the simulation results for different parameterizations with nearly identical shapes of the retention curve. We are aware of the limitations of Mualem's (1976) conductivity curve and of the work of Tuller and Or and others on film flow and corner flow but we are still working on that (see above). The reference to Weber et al. points to a method to implement conductivity curves that are not solely based on capillary flow. As explained above, we will add a brief discussion of the issues raised in this and another comment below Eq. (13).

*Line 161, eq. (5): I understand that the value of parameter beta is determined using eq. (7). I would have expected that beta should depend on the surface area of the soil (determining the amount of adsorbed water). Could the authors please comment on that?*

Reply:
The requirements that the values and the derivatives of two branches of the retention curves match at their junction point were introduced by Rossi and Nimmo (1994). These requirements provide two extra equations that allowed us to solve for two of the variables. The choice of variables for which to solve is arbitrary in principle, but for this particular problem only $\beta$ and $h_d$ can be expressed in an explicit from that does not require an iterative solution. This strictly mathematical line of argument in no way

negates the point of the referee that $\beta$ may depend on the surface area of the soil. Equation (7) and the reasoning of the referee are both correct in our view.

All references appear in the discussion paper.

On behalf of all authors,

Gerrit de Rooij

---

## Author Response (AR1)

**Revisions based on referee comments.**

Line numbers in our replies refer to the version with the marked changes.

Referee 1

Referee 1 wanted more clarity about Eq. (4). We added Rossi and Nimmo's (1994) model as Eq. (4) to the Introduction and discussed it in detail there. Our new parameterization is therefore Eq. (5) in the revision. We added a line below that equation explaining how the different branches were taken from Eqs. (3) and (4). Referee 1 also asked us to list the advantages of Eq. (5), but we already did that in the original text (lines 155-157 of the revision).

We did not change the figures because referee 1 only had problems when the figures were printed in black and white, which is a minor issue.

Referee 2

We expanded the discussion of the unphysical behavior of retention curves that have non-converging integrals as reported by Fuentes et al. (1991) (lines 63-70).

In the Theory section the subsection on the hydraulic conductivity curve, lines have been added to indicate that the main focus on the paper is on the retention curve and the examination of the hydraulic conductivity curve limits itself to finding a closed-form expression. We also explained the limitations of using fixed parameters according to theoretical model when no conductivity data are available and the possibility to fit conductivity parameters independently from the retention curve if the data allow this. We added a line in the Results and Discussion section alluding to this added text (line 384-386).

A brief discussion of the limitations of the conductivity model was added below Eq. (14b). This paragraph concludes with the reference to a recent publication with more sophisticated conductivity models that was already included in the first version.

We decided against seeking additional data to test our parameterization because the hydraulic conductivity is not the main focus of the paper and the measurement technology required to produce the type of data that would be needed does not exist to our knowledge.

Comments not addressed in our earlier response to Referee 2:

*Line 209, eq. (13): Is tau chosen as 0.5?*

The text as written is correct: the expression is valid for any value of $\tau$. See Madi et al. (2018) for limits to its range. When we used the retention parameters to estimate conductivity curves we set the parameter values in Eq. (14a) (Eq. (13a) in the first version) according to Mualem (1976). The text mentions this, and also gives these parameter values, including that of $\tau$.

*Lines 225 – 231: I propose to choose less but better data with (i) consistent lab measurements for both SWRC and K(theta) and (ii) some data with K(theta) values at very negative pressures*

If we could have we would have done so, but to our knowledge there are no conductivity measurements at low matric potentials that do not rely on conversion from water contents to matric potential through the water retention curve. In a paper that introduces a new retention curve parameterization, the reliance of the conductivity data on that same parameterization would create complications in the logic of the line of argumentation.

*Figure 1: Why RIA and VGA are different for the loamy sand?*

This is why we argue for more data points near saturation. There is not enough information in the data to pinpoint the air-entry value, and VGA and RIA have enough flexibility to produce different curves that match the available data very well. The fitting code in one case settled on an air-entry value very close to zero, and in the other case on a value closer to the wettest unsaturated observation point. If the data provide more guidance in the range near-saturation, this will reduce the uncertainty about the shape of the retention curve that is evident from Fig. 1.

*Table 2: You should add Ks and hd (and beta) values for RIA*

The table only contains the fitting parameters, parameters that are expressed as functions of the fitting parameters should therefore not appear in the table. Table 2 is strictly concerned with the retention curve. $K_s$ does not appear in the retention curve.

*Figure 2: Choose different color for VGA for silt loam – it is too similar to clay*

We changed the color to a lighter hue of blue in both Fig. 1 and Fig. 2 for consistency.

*Lines 293 - 298: The reference to Bitelli and Flury is illustrative – maybe the authors could use some of those data as well to fit SWRC*

We added fits to one of Bittelli and Flury's (2009) samples, because all samples were from the same location and the disturbed samples were of limited interest to us. The three sigmoidal parameterizations tested in this paper gave very similar results, except in the very dry range, where RIA's log-linear branch deviated from the asymptotic branches of the other two. The difference between the data sets resulted in distinctly different fitted SWRCs. The results corroborate the rest of the paper, as well as Bittelli and Flury's (2009) findings. Interestingly, the inclusion of data points in the dry range resulted in a value of the matric potential at oven dryness that is very close to the value reported in the literature as being representative for many ovens. We added a section (including a figure and a table) in the Results and Discussion starting at line 350.

*Line 320: the only sample with K(theta) measured in the lab (soil 4450) is poorly described by RIA …*

We modified the text to explain that fitting the conductivity curve with retention data only is not a good idea.

*Lines 331-335: There is no experimental evidence that the RIA trends are better – this is just a description of modeled behavior.*

This is correct. The text reflects this and makes no claims about RIA being better.

*Figure C5 & C9, Soil 1122 (and other examples): The authors obviously cut the curves at pF=6.8 for VGA and VGN. Probably this should be stated and explained in the captions.*

The referee is correct. A sentence explaining this was added to the third paragraph of Appendix C.

*Lines 363-365: The statement that 'small differences between SWRCs can have a significant influence through different hydraulic conductivity curves' should probably be revised; even for the very same SWRC curve the water flow will be different due to different conductivity functions*

Valid point. We clarified the text.

*Lines385-386: The statement that 'RIA was better able to produce a conductivity curve with a substantial drop ...' is misleading because we do not know if this 'substantial drop' is in agreement with measurements*

We used 'produce', not 'reproduce'. The statement therefore is not misleading but an accurate description of the capability of RIA to produce conductivity curves with substantial drops just below saturation. For soils with well-connected pore networks and perhaps tunnel-like biopores (i.e., root holes), such drops can be expected. We therefore posit that this is a desirable property.

TECHNICAL CORRECTIONS/COMMENTS

*Line 17: State that the infinite slope at saturation is considered to be physically impossible*

Done.

*Line 37: Write out SWRC at the beginning of the main text*

Because we use many abbreviations we listed them all in Appendix A. Therefore we did not write them out in full on their first occurrence.

*Line 96: the shape of the samples ('cylindrical') is not relevant; maybe 'equilibrating short soil samples'?*

Because the samples are cylindrical, their cross-sectional area is independent of the sample height. When the fitting code calculates the water content of the sample, it therefore assigns the same weight to each depth interval. Therefore, the shape of the samples is relevant. So is their orientation. We therefore added 'vertically placed'.

*Line 157: delete 'the' ('... the the logarithmic ...)*

Thank you. Corrected.

*Line 238: Is Tamale not in the tropical climate zone?*

No, the climate is semi-arid there. Tamale is located between the tropics to the south and the Sahel to the north.

*Line 315: delete 'goes'*

Corrected.

*Line 344-345: what do you mean with 'reduced Ks-value' and the 'high Ks-value' for clay soils?*

As the text at the end of Material and Methods explains, UNSODA had two samples for the same location with markedly different values for $K_s$. In all probability, one of them had macropores, and the other did not. Because $K_s$ scales the conductivity curve, using the macroporous $K_s$ lifted the rest of the conductivity curve, and the values became unrealistically high for capillary flow in a clay soil. We therefore used the $K_s$ -value of the other sample instead.

Line 352: for the loamy sand under temperate climate, the results ranged from 92 to 104 % (Table 3) – why is this difference more than 10%? Was the deviation in the silt loam (84-115%) not higher?

The referee is right. We corrected the mistake in the text.

*Line 353: For loamy sand in 'semi-arid' climate, the 89% value for evaporation has higher deviation than 10%*

We corrected the text.

*Line 354: Temperate, not 'temperature'*

Corrected.

*Line 439: delete 'of'*

Corrected.

---

## Author Response (AR2)

Reply to review report 2 on the revision. The comment s are in *blue and italic*, our response in black and regular font.

*The revised article including the new figure 3 has improved and is easy to read. Before publication, some technical aspects related to the hydraulic conductivity function must be clarified.*

*1) I tried to reproduce some of the plotted K(theta) curves for RIA and VGA model and did not get the same curves. Possibly/probably, I made a typo in the equations and because of that I have different curves. However, as a reviewer (and potential user) I must be sure how the authors computed the curves.*
*Specifically, for RIA, I implemented eq. (14) using eq. (8) and (9) for beta and h_d, respectively. I tried to compute K(theta) for UNSODA sample 1143 using parameters listed in Table C2. On page 17, lines 298 and 299, it is written "Mualems model for the unsaturated hydraulic conductivity curve was used …" to plot K(theta). So I'm assuming they refer to eq. (14) with gamma=2 and tau = ½. Is that correct?*

Yes

*Given the complexity of the equations, an excel sheet (or a few lines of codes in matlab or python) to enable the user to reproduce a specific example would be helpful (for example as supplementary information file).*

If one first calculates recurring terms involving multiple variables and stores these separately, the full equations can be built with relative ease. The guidelines are not permitting the inclusion of program code as supplemental material. If an exception is granted, the first author will provide the Fortran routines for the various parameterizations used in the paper.

*2) Similarly, I'm not sure how the conductivity function for the VGA curve was computed. Did the authors apply eq. (11) of Ippisch et al. (2006)? As the authors state on page 24, lines 396-399: "Near saturation, RIA's $K(\theta)$ curve often drops off sharply before leveling off, in stark contrast to that of VGA, which remains high in the wet range. Given the similarity in the $\theta(h)$ curves of VGA and RIA, the difference in their $K(\theta)$ curves is remarkable."*

The paper presents only one set of equations (14a and 14b) that provide an explicit $K(h)$ function, so there is no ambiguity about the calculation of $K(h)$ - Comment (1)above correctly refers to these equations. We slightly modified the text below Fig. (1) to clarify this even further.

*The authors do not explore or explain why there is a big difference in K(theta) between VGA and RIA for several cases. It must be possible to explain this from a mathematical point of view. The authors must expand on this (explain the differences between VGA and RIA close to saturation) and must provide the expression for K(theta) they used to plot VGA.*

We explained above which equations were used. If one calculates $\theta$ and $K$ for a range of $h$-values, the $K(\theta)$ data points are obtained as well.

We agree that the differences in the conductivity curve can be explained mathematically, and it is clear from Table 2 that $\alpha$ and $n$ have interestingly different values in some cases. Finding out how exactly these affect the shape of the conductivity curve is quite laborious. In the first round, the referee criticized the limitations of the conductivity curve because it did not account for film flow and corner flow. We explained we are working on that, and prefer to focus our energy on that. It would take several days at least to develop plots of $K(h)$ for different combinations of $\alpha$ and $n$, and we may end up recommending different parameterizations in our future work anyway.

For those who are interested in exploring this, the paper provides all the information they need to get started.

*3) In Appendix C, the authors show a comparison of measured and modeled conductivity data. Only in one case the conductivity values were measured in the lab, for all other cases lab-based SWRC parameters inserted into the conductivity function were compared to field measurements (see page 23/24, line 386-388). From my point of view, such a comparison is meaningless and does not help to understand which K(theta) model gives a more realistic description – the results are just dominated by different sample sizes and structures (as the authors know). A more theoretically based description of different K(theta) shapes based on the SWRC parameters would be more insightful.*

This comment was raised in the first round, and we refer to our earlier reply.

*Another aspect that the authors should explain in more detail is the value of h_d given in eq. (9) (matric potential at which water content reaches zero). Should this value not be universal (as the authors discuss with pF=6.8)? Is this not just the potential to withdraw the last monolayer of water around the particles? How can the authors defend a value of pF = 10.5 as in Figure 1 for clay?*

See our discussion of Figure 3. When insufficient, or incorrect, data are available, physically unrealistic values of $h_d$ will be fitted. Our point of view is that this is not a weakness of the parameterization, but rather an indication that such unrealistic results point to the need to double check the data and/or add additional data points in the dry range. The UNSODA database mostly consists of data that predate Bitelli and Flury (2009). Our work shows that there is a certain risk when these data uses uncritically. Fitted values of $h_d$ can be used as a diagnostic.

The referee appears to imply claim that we defend unrealistically high values of $h_d$. This is patently false. We explicitly state these values are unrealistic, explain the cause of this, and propose a potential remedy.

*MINOR COMMENTS:*

*Page 2, line 29: Rephrase or specify "The new curve was more robust than the other two (for more simulation scenarios, convergence to numerical solution was achieved)"*

The parenthesized section specifies why the new curve was more robust. We do not understand this comment.

*Page 7, lines 133-134: the sentence "Madi et al. ...." is not clear to me. Could the authors specify the constraints and why (and which) usual models are ruled out?*

The derivation of the criterion that helps determine these constraints, the application of this criterion to 18 different parameterizations, and the resulting mathematical definitions of the constraints are fully explained in Madi et al. (2018). The paper is open access and we recommend that the referee and anyone else interested in the analysis consult it. Complying with the referee's request would force us to repeat verbatim a sizeable piece of text from that paper without adding anything new.

*Page 7, lines 141 ff: This paragraph on the hydraulic functions in the wet range is poorly connected to the previous paragraph on dry range. Please rephrase.*

We changed the order of the paragraphs.

*Page 8, line 144: I suggest to list the equations of Brooks and Corey and Fayer and Simmons model as well (for example in the Appendix)*

Why repeat Madi et al. (2018) here? These equations are not needed in the rest of the paper.

*Page 11, lines 206 ff: A main benefit of the multimodal curve is to describe the effect of macropores on SWRC; but equation (10) is only valid with the constraint that (1/alpha_i) is larger than |h_ae|. Is this constraint meaningful? Could the authors please comment on that?*

From discussions about that with colleagues who strongly favor multimodal curves and reading some of their papers it appears that they consider multimodality to be an expression of particularities in the particle size distribution that lead to multimodal distribution of pore sizes below the macropore range. We prefer to simply offer the multimodal version here and let the section of the research community involved in that line of research explore the issue further.

*Page 12, line 224: The authors state that the primary focus is on SWRC – from my point of view the effect of SWRC on K(theta) or K(h) is as important as well ...*

The referee is of course entitled to her or his view, but our focus is on the retention curve. We will probably address the conductivity in a future paper.

*Page 12, line 231: define saturation S_e*

Done.

*Page 12, line 233: must be eq. (13), not eq. (12)*

The referee is right. We corrected the number.

*Page 14, line 265: what do the authors mean with "sufficient"*

For several soils in UNSODA, the range of matric potentials for which retention data are provided is too small to trace the intermediate and the drying branch of the retention curve. These data were of no use for our purpose.

*Page 15, line 276: for the convenience of the reader, give saturated hydraulic conductivity value as well*

We do not understand this comment. Why would this be convenient to the reader? The saturated hydraulic conductivity is only relevant for the simulations, and the values are provided in the supplement that deals with the simulations.

*Figure 1: Explain in the captions why you stop at pF=6.8 for VGA and VGN*

Above, the referee chastised us for reporting $h_d$ values beyond the theoretical maximum of pF 6.8, and here we are required to explain why we stop at this value. We can only generate a finite number of data points from which to make our graphs. For the curves that have a finite value for $h_d$, we made its absolute value the upper limit of the table with retention points. Asymptotic functions have an infinite value for $h_d$, and we therefore chose the cut-off at the theoretical maximum (pF 6.8). We have never come across a simulation that reached this value, so to devote a caption to the choice for a cut-off that is adequate for all reasonable scenarios seems to be a bit over the top.

*Page 19, line 321: The authors write that RIA has clear advantage compared to RNA and FSB – this is a bit misleading or incomplete, because also VGN and VGA have the same advantage (and*

*perform slightly better than RIA)*

This is a strange comment. The sentence the referee refers to is the first of a paragraph in which we compare RIA to all other parameterizations that we fitted. in the following sentences of that paragraph we compare RIA against VGN and VGA.

*Page 28, Summary and conclusions: For the convenience of the reader that are not reading the entire article, explain RIA, VGA and VGN*

That would be against the guidelines for authors. Furthermore, these abbreviations are explained in the list of abbreviations (Appendix A) on the next page.

*Page 32, line 534: Do you refer to eq. (14a) and (14b)?*

Thank you, yes. We corrected the equation numbers.

On behalf of all authors,

Gerrit de Rooij, January 2021